# Cost function dependent barren plateaus in shallow parametrized quantum circuits

M. Cerezo [1,2✉], Akira Sone[1,2], Tyler Volkoff[1], Lukasz Cincio[1] & Patrick J. Coles[1✉]

Variational quantum algorithms (VQAs) optimize the parameters **θ** of a parametrized quantum circuit $V(\boldsymbol{\theta})$ to minimize a cost function $C$. While VQAs may enable practical applications of noisy quantum computers, they are nevertheless heuristic methods with unproven scaling. Here, we rigorously prove two results, assuming $V(\boldsymbol{\theta})$ is an alternating layered ansatz composed of blocks forming local 2-designs. Our first result states that defining $C$ in terms of global observables leads to exponentially vanishing gradients (i.e., barren plateaus) even when $V(\boldsymbol{\theta})$ is shallow. Hence, several VQAs in the literature must revise their proposed costs. On the other hand, our second result states that defining $C$ with local observables leads to at worst a polynomially vanishing gradient, so long as the depth of $V(\boldsymbol{\theta})$ is $\mathcal{O}(\log n)$. Our results establish a connection between locality and trainability. We illustrate these ideas with large-scale simulations, up to 100 qubits, of a quantum autoencoder implementation.

[1] Theoretical Division, Los Alamos National Laboratory, Los Alamos, NM, USA. [2] Center for Nonlinear Studies, Los Alamos National Laboratory, Los Alamos, NM, USA. ✉email: cerezo@lanl.gov; pcoles@lanl.gov

One of the most important technological questions is whether Noisy Intermediate-Scale Quantum (NISQ) computers will have practical applications[1]. NISQ devices are limited both in qubit count and in gate fidelity, hence preventing the use of quantum error correction.

The leading strategy to make use of these devices is variational quantum algorithms (VQAs)[2]. VQAs employ a quantum computer to efficiently evaluate a cost function $C$, while a classical optimizer trains the parameters $\boldsymbol{\theta}$ of a Parametrized Quantum Circuit (PQC) $V(\boldsymbol{\theta})$. The benefits of VQAs are three-fold. First, VQAs allow for task-oriented programming of quantum computers, which is important since designing quantum algorithms is non-intuitive. Second, VQAs make up for small qubit counts by leveraging classical computational power. Third, pushing complexity onto classical computers, while only running short-depth quantum circuits, is an effective strategy for error mitigation on NISQ devices.

There are very few rigorous scaling results for VQAs (with exception of one-layer approximate optimization[3–5]). Ideally, in order to reduce gate overhead that arises when implementing on quantum hardware one would like to employ a hardware-efficient ansatz[6] for $V(\boldsymbol{\theta})$. As recent large-scale implementations for chemistry[7] and optimization[8] applications have shown, this ansatz leads to smaller errors due to hardware noise. However, one of the few known scaling results is that deep versions of randomly initialized hardware-efficient ansatzes lead to exponentially vanishing gradients[9]. Very little is known about the scaling of the gradient in such ansatzes for shallow depths, and it would be especially useful to have a converse bound that guarantees non-exponentially vanishing gradients for certain depths. This motivates our work, where we rigorously investigate the gradient scaling of VQAs as a function of the circuit depth.

The other motivation for our work is the recent explosion in the number of proposed VQAs. The Variational Quantum Eigensolver (VQE) is the most famous VQA. It aims to prepare the ground state of a given Hamiltonian $H = \sum_\alpha c_\alpha \sigma_\alpha$, with $H$ expanded as a sum of local Pauli operators[10]. In VQE, the cost function is obviously the energy $C = \langle \psi | H | \psi \rangle$ of the trial state $|\psi\rangle$. However, VQAs have been proposed for other applications, like quantum data compression[11], quantum error correction[12], quantum metrology[13], quantum compiling[14–17], quantum state diagonalization[18,19], quantum simulation[20–23], fidelity estimation[24], unsampling[25], consistent histories[26], and linear systems[27–29]. For these applications, the choice of $C$ is less obvious. Put another way, if one reformulates these VQAs as ground-state problems (which can be done in many cases), the choice of Hamiltonian $H$ is less intuitive. This is because many of these applications are abstract, rather than associated with a physical Hamiltonian.

We remark that polynomially vanishing gradients imply that the number of shots needed to estimate the gradient should grow as $\mathcal{O}(\text{poly}(n))$. In contrast, exponentially vanishing gradients (i.e., barren plateaus) imply that derivative-based optimization will have exponential scaling[30], and this scaling can also apply to derivative-free optimization[31]. Assuming a polynomial number of shots per optimization step, one will be able to resolve against finite sampling noise and train the parameters if the gradients vanish polynomially. Hence, we employ the term "trainable" for polynomially vanishing gradients.

In this work, we connect the trainability of VQAs to the choice of $C$. For the abstract applications in refs. [11–29], it is important for $C$ to be operational, so that small values of $C$ imply that the task is almost accomplished. Consider an example of state preparation, where the goal is to find a gate sequence that prepares a target state $|\psi_0\rangle$. A natural cost function is the square of the trace distance $D_T$ between $|\psi_0\rangle$ and $|\psi\rangle = V(\boldsymbol{\theta})^\dagger |\boldsymbol{0}\rangle$, given by $C_G = D_T(|\psi_0\rangle, |\psi\rangle)^2$, which is equivalent to

$$C_G = \text{Tr}[O_G V(\boldsymbol{\theta}) |\psi_0\rangle \langle\psi_0| V(\boldsymbol{\theta})^\dagger], \qquad (1)$$

with $O_G = \mathbb{1} - |\boldsymbol{0}\rangle\langle\boldsymbol{0}|$. Note that $\sqrt{C_G} \geq |\langle\psi|M|\psi\rangle - \langle\psi_0|M|\psi_0\rangle|$ has a nice operational meaning as a bound on the expectation value difference for a POVM element $M$.

However, here we argue that this cost function and others like it exhibit exponentially vanishing gradients. Namely, we consider global cost functions, where one directly compares states or operators living in exponentially large Hilbert spaces (e.g., $|\psi\rangle$ and $|\psi_0\rangle$). These are precisely the cost functions that have operational meanings for tasks of interest, including all tasks in refs. [11–29]. Hence, our results imply that a non-trivial subset of these references will need to revise their choice of $C$.

Interestingly, we demonstrate vanishing gradients for shallow PQCs. This is in contrast to McClean et al.[9], who showed vanishing gradients for deep PQCs. They noted that randomly initializing $\boldsymbol{\theta}$ for a $V(\boldsymbol{\theta})$ that forms a 2-design leads to a barren plateau, i.e., with the gradient vanishing exponentially in the number of qubits, $n$. Their work implied that researchers must develop either clever parameter initialization strategies[32,33] or clever PQCs ansatzes[4,34,35]. Similarly, our work implies that researchers must carefully weigh the balance between trainability and operational relevance when choosing $C$.

While our work is for general VQAs, barren plateaus for global cost functions were noted for specific VQAs and for a very specific tensor-product example by our research group[14,18], and more recently in[29]. This motivated the proposal of local cost functions[14,16,18,22,25–27], where one compares objects (states or operators) with respect to each individual qubit, rather than in a global sense, and therein it was shown that these local cost functions have indirect operational meaning.

Our second result is that these local cost functions have gradients that vanish polynomially rather than exponentially in $n$, and hence have the potential to be trained. This holds for $V(\boldsymbol{\theta})$ with depth $\mathcal{O}(\log n)$. Figure 1 summarizes our two main results.

Finally, we illustrate our main results for an important example: quantum autoencoders[11]. Our large-scale numerics show that the global cost function proposed in[11] has a barren plateau. On the other hand, we propose a novel local cost function that is trainable, hence making quantum autoencoders a scalable application.

## Results

**Warm-up example.** To illustrate cost-function-dependent barren plateaus, we first consider a toy problem corresponding to the state preparation problem in the Introduction with the target state being $|\boldsymbol{0}\rangle$. We assume a tensor-product ansatz of the form $V(\boldsymbol{\theta}) = \bigotimes_{j=1}^n e^{-i\theta^j \sigma_x^{(j)}/2}$, with the goal of finding the angles $\theta^j$ such that $V(\boldsymbol{\theta})|\boldsymbol{0}\rangle = |\boldsymbol{0}\rangle$. Employing the global cost of (1) results in $C_G = 1 - \prod_{j=1}^n \cos^2 \frac{\theta^j}{2}$. The barren plateau can be detected via the variance of its gradient: $\text{Var}[\frac{\partial C_G}{\partial \theta^j}] = \frac{1}{8}(\frac{3}{8})^{n-1}$, which is exponentially vanishing in $n$. Since the mean value is $\langle \frac{\partial C_G}{\partial \theta^j} \rangle = 0$, the gradient concentrates exponentially around zero.

On the other hand, consider a local cost function:

$$C_L = \text{Tr}\left[O_L V(\boldsymbol{\theta})|\boldsymbol{0}\rangle \langle\boldsymbol{0}| V(\boldsymbol{\theta})^\dagger\right], \qquad (2)$$

$$\text{with} \quad O_L = \mathbb{1} - \frac{1}{n}\sum_{j=1}^n |0\rangle\langle 0|_j \otimes \mathbb{1}_{\bar{j}}, \qquad (3)$$

where $\mathbb{1}_{\bar{j}}$ is the identity on all qubits except qubit $j$. Note that $C_L$

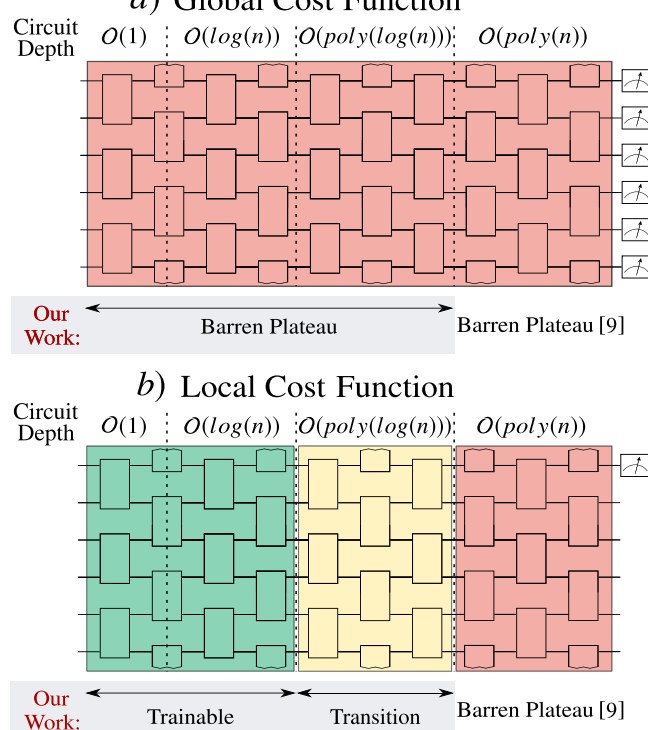

**Fig. 1 Summary of our main results.** McClean et al.[9] proved that a barren plateau can occur when the depth $D$ of a hardware-efficient ansatz is $D \in \mathcal{O}(\mathrm{poly}\,(n))$. Here we extend these results by providing bounds for the variance of the gradient of global and local cost functions as a function of $D$. In particular, we find that the barren plateau phenomenon is cost-function dependent. **a** For global cost functions (e.g., Eq. (1)), the landscape will exhibit a barren plateau essentially for all depths $D$. **b** For local cost functions (e.g., Eq. (2)), the gradient vanishes at worst polynomially and hence is trainable when $D \in \mathcal{O}(\log\,(n))$, while barren plateaus occur for $D \in \mathcal{O}(\mathrm{poly}\,(n))$, and between these two regions the gradient transitions from polynomial to exponential decay.

vanishes under the same conditions as $C_{\mathrm{G}}$[14,16], $C_{\mathrm{L}} = 0 \Leftrightarrow C_{\mathrm{G}} = 0$. We find $C_{\mathrm{L}} = 1 - \frac{1}{n}\sum_{j=1}^{n}\cos^2\frac{\theta^j}{2}$, and the variance of its gradient is $\mathrm{Var}\left[\frac{\partial C_{\mathrm{L}}}{\partial \theta^j}\right] = \frac{1}{8n^2}$, which vanishes polynomially with $n$ and hence exhibits no barren plateau. Figure 2 depicts the cost landscapes of $C_{\mathrm{G}}$ and $C_{\mathrm{L}}$ for two values of $n$ and shows that the barren plateau can be avoided here via a local cost function.

Moreover, this example allows us to delve deeper into the cost landscape to see a phenomenon that we refer to as a narrow gorge. While a barren plateau is associated with a flat landscape, a narrow gorge refers to the steepness of the valley that contains the global minimum. This phenomenon is illustrated in Fig. 2, where each dot corresponds to cost values obtained from randomly selected parameters $\boldsymbol{\theta}$. For $C_{\mathrm{G}}$ we see that very few dots fall inside the narrow gorge, while for $C_{\mathrm{L}}$ the narrow gorge is not present. Note that the narrow gorge makes it harder to train $C_{\mathrm{G}}$ since the learning rate of descent-based optimization algorithms must be exponentially small in order not to overstep the narrow gorge. The following proposition (proved in the Supplementary Note 2) formalizes the narrow gorge for $C_{\mathrm{G}}$ and its absence for $C_{\mathrm{L}}$ by characterizing the dependence on $n$ of the probability $C \leqslant \delta$. This probability is associated with the parameter space volume that leads to $C \leqslant \delta$.

**Proposition 1**
Let $\theta^j$ be uniformly distributed on $[-\pi, \pi]\ \forall j$. For any $\delta \in (0, 1)$, the probability that $C_{\mathrm{G}} \leq \delta$ satisfies

$$\Pr\{C_{\mathrm{G}} \leq \delta\} \leq (1 - \delta)^{-1}\left(\frac{1}{2}\right)^n . \qquad (4)$$

For any $\delta \in [\frac{1}{2}, 1]$, the probability that $C_{\mathrm{L}} \leq \delta$ satisfies

$$\Pr\{C_{\mathrm{L}} \leq \delta\} \geq \frac{(2\delta - 1)^2}{\frac{1}{2n} + (2\delta - 1)^2} \xrightarrow[n\to\infty]{} 1 . \qquad (5)$$

**General framework**. For our general results, we consider a family of cost functions that can be expressed as the expectation value of an operator $O$ as follows

$$C = \mathrm{Tr}\left[OV(\boldsymbol{\theta})\rho V^{\dagger}(\boldsymbol{\theta})\right] , \qquad (6)$$

where $\rho$ is an arbitrary quantum state on $n$ qubits. Note that this framework includes the special case where $\rho$ could be a pure state, as well as the more special case where $\rho = |\mathbf{0}\rangle\langle\mathbf{0}|$, which is the input state for many VQAs such as VQE. Moreover, in VQE, one chooses $O = H$, where $H$ is the physical Hamiltonian. In general, the choice of $O$ and $\rho$ essentially defines the application of interest of the particular VQA.

It is typical to express $O$ as a linear combination of the form $O = c_0 \mathbb{1} + \sum_{i=1}^{N} c_i O_i$. Here $O_i \neq \mathbb{1}$, $c_i \in \mathbb{R}$, and we assume that at least one $c_i \neq 0$. Note that $C_{\mathrm{G}}$ and $C_{\mathrm{L}}$ in (1) and (2) fall under this framework. In our main results below, we will consider two different choices of $O$ that respectively capture our general notions of global and local cost functions and also generalize the aforementioned $C_{\mathrm{G}}$ and $C_{\mathrm{L}}$.

As shown in Fig. 3a, $V(\boldsymbol{\theta})$ consists of $L$ layers of $m$-qubit unitaries $W_{kl}(\boldsymbol{\theta}_{kl})$, or blocks, acting on alternating groups of $m$ neighboring qubits. We refer to this as an Alternating Layered Ansatz. We remark that the Alternating Layered Ansatz will be a hardware-efficient ansatz so long as the gates that compose each block are taken from a set of gates native to a specific device. As depicted in Fig. 3c, the one dimensional Alternating Layered Ansatz can be readily implemented in devices with one-dimensional connectivity, as well as in devices with two-dimensional connectivity (such as that of IBM's[36] and Google's[37] quantum devices). That is, with both one- and two-dimensional hardware connectivity one can group qubits to form an Alternating Layered Ansatz as in Fig. 3a.

The index $l = 1, ..., L$ in $W_{kl}(\boldsymbol{\theta}_{kl})$ indicates the layer that contains the block, while $k = 1, ..., \xi$ indicates the qubits it acts upon. We assume $n$ is a multiple of $m$, with $n = m\xi$, and that $m$ does not scale with $n$. As depicted in Fig. 3a, we define $S_k$ as the $m$-qubit subsystem on which $W_{kL}$ acts, and we define $\mathcal{S} = \{S_k\}$ as the set of all such subsystems. Let us now consider a block $W_{kl}(\boldsymbol{\theta}_{kl})$ in the $l$th layer of the ansatz. For simplicity we henceforth use $W$ to refer to a given $W_{kl}(\boldsymbol{\theta}_{kl})$. As shown in the Methods section, given a $\theta^v \in \boldsymbol{\theta}_{kl}$ that parametrizes a rotation $e^{-i\theta^v\sigma_v/2}$ (with $\sigma_v$ a Pauli operator) inside a given block $W$, one can always express

$$\frac{\partial W}{\partial \theta^v} := \partial_v W = \frac{-i}{2} W_{\mathrm{A}}\sigma_v W_{\mathrm{B}}, \qquad (7)$$

where $W_{\mathrm{A}}$ and $W_{\mathrm{B}}$ contain all remaining gates in $W$, and are properly defined in the Methods section.

The contribution to the gradient $\nabla C$ from a parameter $\theta^v$ in the block $W$ is given by the partial derivative $\partial_v C$. While the value of $\partial_v C$ depends on the specific parameters $\boldsymbol{\theta}$, it is useful to compute $\langle\partial_v C\rangle_V$, i.e., the average gradient over all possible unitaries $V(\boldsymbol{\theta})$ within the ansatz. Such an average may not be representative near

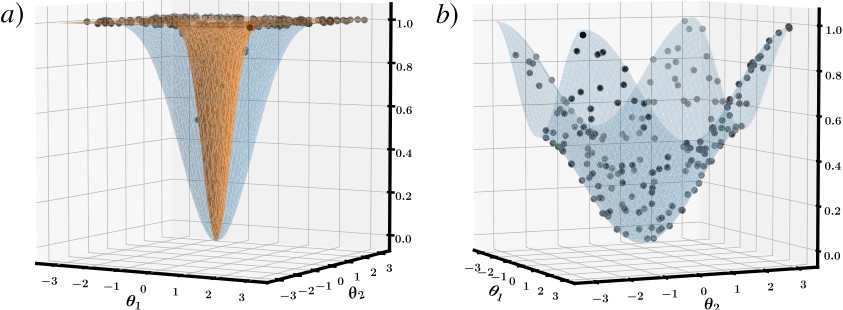

**Fig. 2 Cost function landscapes. a** Two-dimensional cross-section through the landscape of $C_G = 1 - \prod_{j=1}^{n} \cos^2(\theta^j/2)$ for $n = 4$ (blue) and $n = 24$ (orange). **b** The same cross-section through the landscape of $C_L = 1 - \frac{1}{n}\sum_{j=1}^{n} \cos^2(\theta^j/2)$ is independent of $n$. In both cases, 200 Haar distributed points are shown, with very few (most) of these points lying in the valley containing the global minimum of $C_G$ ($C_L$).

the minimum of $C$, although it does provide a good estimate of the expected gradient when randomly initializing the angles in $V(\boldsymbol{\theta})$. In the Methods Section we explicitly show how to compute averages of the form $\langle \ldots \rangle_V$, and in the Supplementary Note 3 we provide a proof for the following Proposition.

**Proposition 2**

The average of the partial derivative of any cost function of the form (6) with respect to a parameter $\theta^v$ in a block $W$ of the ansatz in Fig. 3 is

$$\langle \partial_v C \rangle_V = 0 , \qquad (8)$$

provided that either $W_A$ or $W_B$ of (7) form a 1-design.

Here we recall that a $t$-design is an ensemble of unitaries, such that sampling over their distribution yields the same properties as sampling random unitaries from the unitary group with respect to the Haar measure up to the first $t$ moments[38]. The Methods section provides a formal definition of a $t$-design.

Proposition 2 states that the gradient is not biased in any particular direction. To analyze the trainability of $C$, we consider the second moment of its partial derivatives:

$$\mathrm{Var}[\partial_v C] = \langle (\partial_v C)^2 \rangle_V , \qquad (9)$$

where we used the fact that $\langle \partial_v C \rangle_V = 0$. The magnitude of $\mathrm{Var}[\partial_v C]$ quantifies how much the partial derivative concentrates around zero, and hence small values in (9) imply that the slope of the landscape will typically be insufficient to provide a cost-minimizing direction. Specifically, from Chebyshev's inequality, $\mathrm{Var}[\partial_v C]$ bounds the probability that the cost-function partial derivative deviates from its mean value (of zero) as $\mathrm{Pr}(|\partial_v C| \geq c) \leq \mathrm{Var}[\partial_v C]/c^2$ for all $c > 0$.

**Main results**. Here we present our main theorems and corollaries, with the proofs sketched in the Methods and detailed in the Supplementary Information. In addition, in the Methods section we provide some intuition behind our main results by analyzing a generalization of the warm-up example where $V(\boldsymbol{\theta})$ is composed of a single layer of the ansatz in Fig. 3. This case bridges the gap between the warm-up example and our main theorems and also showcases the tools used to derive our main result.

The following theorem provides an upper bound on the variance of the partial derivative of a global cost function which can be expressed as the expectation value of an operator of the form

$$O = c_0 \mathbb{1} + \sum_{i=1}^{N} c_i \widehat{O}_{i1} \otimes \widehat{O}_{i2} \otimes \cdots \otimes \widehat{O}_{i\xi} . \qquad (10)$$

Specifically, we consider two cases of interest: (i) When $N = 1$ and each $\widehat{O}_{1k}$ is a non-trivial projector ($\widehat{O}_{1k}^2 = \widehat{O}_{1k} \neq \mathbb{1}$) of rank $r_k$

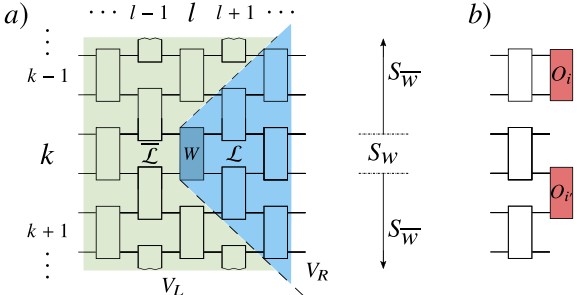

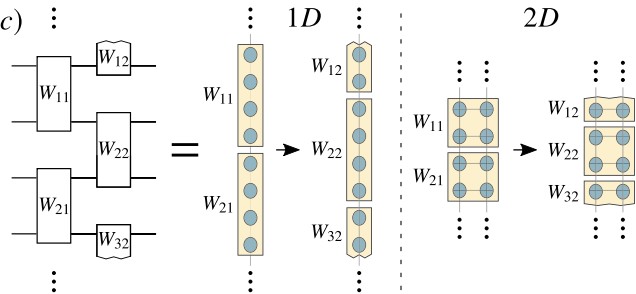

**Fig. 3 Alternating Layered Ansatz. a** Each block $W_{kl}$ acts on $m$ qubits and is parametrized via (27). As shown, we define $S_k$ as the $m$-qubit subsystem on which $W_{kL}$ acts, where $L$ is the last layer of $V(\boldsymbol{\theta})$. Given some block $W$, it is useful for our proofs (outlined in the Methods) to write $V(\boldsymbol{\theta}) = V_R (\mathbb{1}_{\overline{w}} \otimes W)V_L$, where $V_R$ contains all gates in the forward light-cone $\mathcal{L}$ of $W$. The forward light-cone $\mathcal{L}$ is defined as all gates with at least one input qubit causally connected to the output qubits of $W$. We define $\overline{\mathcal{L}}$ as the compliment of $\mathcal{L}$, $S_w$ as the $m$-qubit subsystem on which $W$ acts, and $S_{\overline{w}}$ as the $n - m$ qubit subsystem on which $W$ acts trivially. **b** The operator $O_i$ acts nontrivially only in subsystem $S_{k-1} \in \mathcal{S}$, while $O_{i'}$ acts nontrivially on the first $m/2$ qubits of $S_{k+1}$, and on the second $m/2$ qubits of $S_k$. **c** Depiction of the Alternating Layered Ansatz with one- and two-dimensional connectivity. Each circle represents a physical qubit.

acting on subsystem $S_k$, or (ii) When $N$ is arbitrary and $\widehat{O}_{ik}$ is traceless with $\mathrm{Tr}[\widehat{O}_{ik}^2] \leq 2^m$ (for example, when $\widehat{O}_{ik} = \bigotimes_{j=1}^{m} \sigma_j^\mu$ is a tensor product of Pauli operators $\sigma_j^\mu \in \{\mathbb{1}_j, \sigma_j^x, \sigma_j^y, \sigma_j^z\}$, with at least one $\sigma_j^\mu \neq \mathbb{1}$). Note that case (i) includes $C_G$ of (1) as a special case.

**Theorem 1**

Consider a trainable parameter $\theta^v$ in a block $W$ of the ansatz in Fig. 3. Let $\mathrm{Var}[\partial_v C]$ be the variance of the partial derivative of a global cost function $C$ (with $O$ given by (10)) with respect to $\theta^v$. If

$W_A$, $W_B$ of (7), and each block in $V(\boldsymbol{\theta})$ form a local 2-design, then $\mathrm{Var}[\partial_\nu C]$ is upper bounded by

$$\mathrm{Var}[\partial_\nu C] \leqslant F_n(L, l) \ . \qquad (11)$$

(i) For $N = 1$ and when each $\widehat{O}_{1k}$ is a non-trivial projector, then defining $R = \prod_{k=1}^{\xi} r_k^2$, we have

$$F_n(L, l) = \frac{2^{2m+(2m-1)(L-l)}}{(2^{2m}-1) \cdot 3^{\frac{n}{m}} \cdot 2^{(2-\frac{3}{m})n}} c_1^2 R \ . \qquad (12)$$

(ii) For arbitrary $N$ and when each $\widehat{O}_{ik}$ satisfies $\mathrm{Tr}[\widehat{O}_{ik}] = 0$ and $\mathrm{Tr}[\widehat{O}_{ik}^2] \leqslant 2^m$, then

$$F_n(L, l) = \frac{2^{2m(L-l+1)+1}}{3^{\frac{2n}{m}} \cdot 2^{(3-\frac{4}{m})n}} \sum_{i,j=1}^{N} c_i c_j \ . \qquad (13)$$

From Theorem 1 we derive the following corollary.

## Corollary 1

Consider the function $F_n(L, l)$.

(i) Let $N = 1$ and let each $\widehat{O}_{1k}$ be a non-trivial projector, as in case (i) of Theorem 1. If $c_1^2 R \in \mathcal{O}(2^n)$ and if the number of layers $L \in \mathcal{O}(\mathrm{poly}(\log(n)))$, then

$$F_n(L, l) \in \mathcal{O}\left(2^{-(1-\frac{1}{m}\log_2 3)n}\right) , \qquad (14)$$

which implies that $\mathrm{Var}[\partial_\nu C]$ is exponentially vanishing in $n$ if $m \geqslant 2$.

(ii) Let $N$ be arbitrary, and let each $\widehat{O}_{ik}$ satisfy $\mathrm{Tr}[\widehat{O}_{ik}] = 0$ and $\mathrm{Tr}[\widehat{O}_{ik}^2] \leqslant 2^m$, as in case (ii) of Theorem 1. If $N \in \mathcal{O}(2^n)$, $c_i \in \mathcal{O}(1)$, and if the number of layers $L \in \mathcal{O}(\mathrm{poly}(\log(n)))$, then

$$F_n(L, l) \in \mathcal{O}\left(\frac{1}{2^{(1-\frac{1}{m})n}}\right) , \qquad (15)$$

which implies that $\mathrm{Var}[\partial_\nu C]$ is exponentially vanishing in $n$ if $m \geqslant 2$.

Let us now make several important remarks. First, note that part (i) of Corollary 1 includes as a particular example the cost function $C_G$ of (1). Second, part (ii) of this corollary also includes as particular examples operators with $N \in \mathcal{O}(1)$, as well as $N \in \mathcal{O}(\mathrm{poly}(n))$. Finally, we remark that $F_n(L, l)$ becomes trivial when the number of layers $L$ is $\Omega(\mathrm{poly}(n))$, however, as we discuss below, we can still find that $\mathrm{Var}[\partial_\nu C_G]$ vanishes exponentially in this case.

Our second main theorem shows that barren plateaus can be avoided for shallow circuits by employing local cost functions. Here we consider $m$-local cost functions where each $\widehat{O}_i$ acts nontrivially on at most $m$ qubits and (on these qubits) can be expressed as $\widehat{O}_i = \widehat{O}_i^{\mu_i} \otimes \widehat{O}_i^{\mu'}$:

$$O = c_0 \mathbb{1} + \sum_{i=1}^{N} c_i \widehat{O}_i^{\mu_i} \otimes \widehat{O}_i^{\mu'} \ , \qquad (16)$$

where $\widehat{O}_i^{\mu_i}$ are operators acting on $m/2$ qubits which can be written as a tensor product of Pauli operators. Here, we assume the summation in Eq. (16) includes two possible cases as schematically shown in Fig. 3b: First, when $\widehat{O}_i^{\mu_i}$ ($\widehat{O}_i^{\mu'}$) acts on the first (last) $m/2$ qubits of a given $S_k$, and second, when $\widehat{O}_i^{\mu_i}$ ($\widehat{O}_i^{\mu'}$) acts on the last (first) $m/2$ qubits of a given $S_k$ ($S_{k+1}$). This type of cost function includes any ultralocal cost function (i.e., where the $\widehat{O}_i$ are one-body) as in (2), and also VQE Hamiltonians with up to $m/2$ neighbor interactions. Then, the following theorem holds.

## Theorem 2

Consider a trainable parameter $\theta^\nu$ in a block $W$ of the ansatz in Fig. 3. Let $\mathrm{Var}[\partial_\nu C]$ be the variance of the partial derivative of an $m$-local cost function $C$ (with $O$ given by (16)) with respect to $\theta^\nu$. $W_A$, $W_B$ of (7), and each block in $V(\boldsymbol{\theta})$ form a local 2-design, then $\mathrm{Var}[\partial_\nu C]$ is lower bounded by

$$G_n(L, l) \leqslant \mathrm{Var}[\partial_\nu C] \ , \qquad (17)$$

with

$$\begin{aligned} G_n(L, l) = \ & \frac{2^{m(l+1)-1}}{(2^{2m}-1)^2 (2^{2m}+1)^{L+l}} \\ & \times \sum_{i \in i_{\mathcal{L}}} \sum_{\substack{(k,k') \in k_{\mathcal{L}_B} \\ k' \geqslant k}} c_i^2 \epsilon(\rho_{k,k'}) \epsilon(\widehat{O}_i) \ , \end{aligned} \qquad (18)$$

where $i_{\mathcal{L}}$ is the set of $i$ indices whose associated operators $\widehat{O}_i$ act on qubits in the forward light-cone $\mathcal{L}$ of $W$, and $k_{\mathcal{L}_B}$ is the set of $k$ indices whose associated subsystems $S_k$ are in the backward light-cone $\mathcal{L}_B$ of $W$. Here we defined the function $\epsilon(M) = D_{\mathrm{HS}}(M, \mathrm{Tr}(M)\mathbb{1}/d_M)$ where $D_{\mathrm{HS}}$ is the Hilbert–Schmidt distance and $d_M$ is the dimension of the matrix $M$. In addition, $\rho_{k,k'}$ is the partial trace of the input state $\rho$ down to the subsystems $S_k S_{k+1} \dots S_{k'}$.

Let us make a few remarks. First, note that the $\epsilon(\widehat{O}_i)$ in the lower bound indicates that training $V(\boldsymbol{\theta})$ is easier when $\widehat{O}_i$ is far from the identity. Second, the presence of $\epsilon(\rho_{k,k'})$ in $G_n(L, l)$ implies that we have no guarantee on the trainability of a parameter $\theta^\nu$ in $W$ if $\rho$ is maximally mixed on the qubits in the backwards light-cone.

From Theorem 2 we derive the following corollary for $m$-local cost functions, which guarantees the trainability of the ansatz for shallow circuits.

## Corollary 2

Consider the function $F_n(L, l)$. Let $O$ be an operator of the form (16), as in Theorem 2. If at least one term $c_i^2 \epsilon(\rho_{k,k'}) \epsilon(\widehat{O}_i)$ in the sum in (18) vanishes no faster than $\Omega(1/\mathrm{poly}(n))$, and if the number of layers $L$ is $\mathcal{O}(\log(n))$, then

$$G_n(L, l) \in \Omega\left(\frac{1}{\mathrm{poly}(n)}\right) . \qquad (19)$$

On the other hand, if at least one term $c_i^2 \epsilon(\rho_{k,k'}) \epsilon(\widehat{O}_i)$ in the sum in (18) vanishes no faster than $\Omega(1/2^{\mathrm{poly}(\log(n))})$, and if the number of layers is $\mathcal{O}(\mathrm{poly}(\log(n)))$, then

$$G_n(L, l) \in \Omega\left(\frac{1}{2^{\mathrm{poly}(\log(n))}}\right) . \qquad (20)$$

Hence, when $L$ is $\mathcal{O}(\mathrm{poly}(\log(n)))$ there is a transition region where the lower bound vanishes faster than polynomially, but slower than exponentially.

We finally justify the assumption of each block being a local 2-design from the fact that shallow circuit depths lead to such local 2-designs. Namely, it has been shown that one-dimensional 2-designs have efficient quantum circuit descriptions, requiring $\mathcal{O}(m^2)$ gates to be exactly implemented[38], or $\mathcal{O}(m)$ to be approximately implemented[39,40]. Hence, an $L$-layered ansatz in which each block forms a 2-design can be exactly implemented with a depth $D \in \mathcal{O}(m^2 L)$, and approximately implemented with $D \in \mathcal{O}(mL)$. For the case of two-dimensional connectivity, it has been shown that approximate 2-designs require a circuit depth of $\mathcal{O}(\sqrt{m})$ to be implemented[40]. Therefore, in this case the depth of the layered ansatz is $D \in \mathcal{O}(\sqrt{m}L)$. The latter shows that increasing the dimensionality of the circuit reduces the circuit depth needed to make each block a 2-design.

Moreover, it has been shown that the Alternating Layered Ansatz of Fig. 3 will form an approximate one-dimensional 2-design on $n$ qubits if the number of layers is $\mathcal{O}(n)$[40]. Hence, for deep circuits, our ansatz behaves like a random circuit and we recover the barren plateau result of[9] for both local and global cost functions.

**Numerical simulations.** As an important example to illustrate the cost-function-dependent barren plateau phenomenon, we consider quantum autoencoders[11,41–44]. In particular, the pioneering VQA proposed in ref. [11] has received significant literature attention, due to its importance to quantum machine learning and quantum data compression. Let us briefly explain the algorithm in ref. [11].

Consider a bipartite quantum system $AB$ composed of $n_A$ and $n_B$ qubits, respectively, and let $\{p_\mu, |\psi_\mu\rangle\}$ be an ensemble of pure states on $AB$. The goal of the quantum autoencoder is to train a gate sequence $V(\boldsymbol{\theta})$ to compress this ensemble into the $A$ subsystem, such that one can recover each state $|\psi_\mu\rangle$ with high fidelity from the information in subsystem $A$. One can think of $B$ as the "trash" since it is discarded after the action of $V(\boldsymbol{\theta})$.

To quantify the degree of data compression, ref. [11] proposed a cost function of the form:

$$C_G' = 1 - \text{Tr}[|\boldsymbol{0}\rangle\langle\boldsymbol{0}|\rho_B^{\text{out}}] \qquad (21)$$

$$= \text{Tr}[O_G' V(\boldsymbol{\theta})\rho_{AB}^{\text{in}} V(\boldsymbol{\theta})^\dagger] , \qquad (22)$$

where $\rho_{AB}^{\text{in}} = \sum_\mu p_\mu |\psi_\mu\rangle\langle\psi_\mu|$ is the ensemble-average input state, $\rho_B^{\text{out}} = \sum_\mu p_\mu \text{Tr}_A[|\psi'\rangle\langle\psi'|]$ is the ensemble-average trash state, and $|\psi'\rangle = V(\boldsymbol{\theta})|\psi_\mu\rangle$. Equation (22) makes it clear that $C_G'$ has the form in (6), and $O_G' = \mathbb{1}_{AB} - \mathbb{1}_A \otimes |\boldsymbol{0}\rangle\langle\boldsymbol{0}|$ is a global observable of the form in (10). Hence, according to Corollary 1, $C_G'$ exhibits a barren plateau for large $n_B$. (Specifically, Corollary 1 applies in this context when $n_A < n_B$). As a result, large-scale data compression, where one is interested in discarding large numbers of qubits, will not be possible with $C_G'$.

To address this issue, we propose the following local cost function

$$C_L' = 1 - \frac{1}{n_B}\sum_{j=1}^{n_B}\text{Tr}\left[\left(|0\rangle\langle0|_j \otimes \mathbb{1}_{\bar{j}}\right)\rho_B^{\text{out}}\right] \qquad (23)$$

$$= \text{Tr}[O_L' V(\boldsymbol{\theta})\rho_{AB}^{\text{in}} V(\boldsymbol{\theta})^\dagger] , \qquad (24)$$

where $O_L' = \mathbb{1}_{AB} - \frac{1}{n_B}\sum_{j=1}^{n_B}\mathbb{1}_A \otimes |0\rangle\langle0|_j \otimes \mathbb{1}_{\bar{j}}$, and $\mathbb{1}_{\bar{j}}$ is the identity on all qubits in $B$ except the $j$th qubit. As shown in the Supplementary Note 9, $C_L'$ satisfies $C_L' \leqslant C_G' \leqslant n_B C_L'$, which implies that $C_L'$ is faithful (vanishing under the same conditions as $C_G'$). Furthermore, note that $O_L'$ has the form in (16). Hence Corollary 2 implies that $C_L'$ does not exhibit a barren plateau for shallow ansatzes.

Here we simulate the autoencoder algorithm to solve a simple problem where $n_A = 1$, and where the input state ensemble $\{p_\mu, |\psi_\mu\rangle\}$ is given by

$$|\psi_1\rangle = |0\rangle_A \otimes |0, 0, 0, \ldots, 0\rangle_B , \quad \text{with} \quad p_1 = 2/3 , \qquad (25)$$

$$|\psi_2\rangle = |1\rangle_A \otimes |1, 1, 0, \ldots, 0\rangle_B , \quad \text{with} \quad p_2 = 1/3 . \qquad (26)$$

In order to analyze the cost-function-dependent barren plateau phenomenon, the dimension of subsystem $B$ is gradually increased as $n_B = 10, 15, \ldots, 100$.

**Numerical results.** In our heuristics, the gate sequence $V(\boldsymbol{\theta})$ is given by two layers of the ansatz in Fig. 4, so that the number of gates and parameters in $V(\boldsymbol{\theta})$ increases linearly with $n_B$. Note that

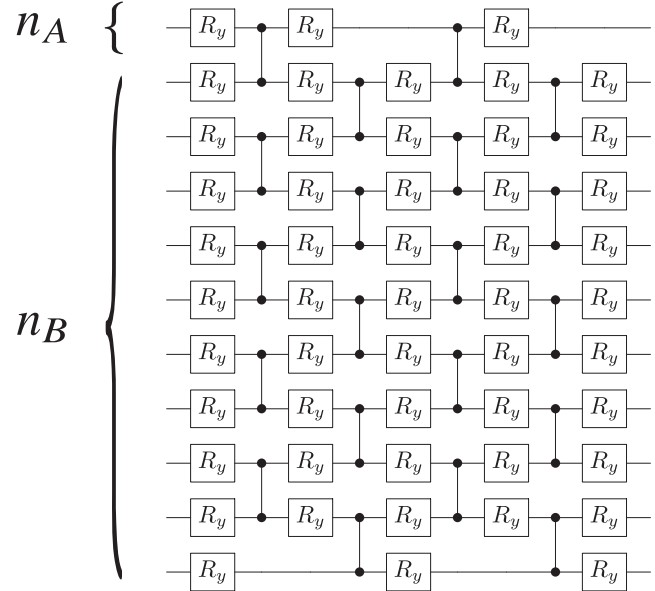

**Fig. 4 Alternating Layered Ansatz for $V(\theta)$ employed in our numerical simulations.** Each layer is composed of control-$Z$ gates acting on alternating pairs of neighboring qubits which are preceded and followed by single qubit rotations around the $y$-axis, $R_y(\theta_i) = e^{-i\theta_i\sigma_y/2}$. Shown is the case of two layers, $n_A = 1$, and $n_B = 10$ qubits. The number of variational parameters and gates scales linearly with $n_B$: for the case shown there are 71 gates and 51 parameters. While each block in this ansatz will not form an exact local 2-design, and hence does not fall under our theorems, one can still obtain a cost-function-dependent barren plateau.

this ansatz is a simplified version of the ansatz in Fig. 3, as we can only generate unitaries with real coefficients. All parameters in $V(\boldsymbol{\theta})$ were randomly initialized and as detailed in the Methods section, we employ a gradient-free training algorithm that gradually increases the number of shots per cost-function evaluation.

Analysis of the $n$-dependence. Figure 5 shows representative results of our numerical implementations of the quantum autoencoder in ref. [11] obtained by training $V(\boldsymbol{\theta})$ with the global and local cost functions respectively given by (22) and (23). Specifically, while we train with finite sampling, in the figures we show the exact cost-function values versus the number of iterations. Here, the top (bottom) axis corresponds to the number of iterations performed while training with $C_G'$ ($C_L'$). For $n_B = 10$ and 15, Fig. 5 shows that we are able to train $V(\boldsymbol{\theta})$ for both cost functions. For $n_B = 20$, the global cost function initially presents a plateau in which the optimizing algorithm is not able to determine a minimizing direction. However, as the number of shots per function evaluation increases, one can eventually minimize $C_G'$. Such result indicates the presence of a barren plateau where the gradient takes small values which can only be detected when the number of shots becomes sufficiently large. In this particular example, one is able to start training at around 140 iterations.

When $n_B > 20$ we are unable to train the global cost function, while always being able to train our proposed local cost function. Note that the number of iterations is different for $C_G'$ and $C_L'$, as for the global cost function case we reach the maximum number of shots in fewer iterations. These results indicate that the global cost function of (22) exhibits a barren plateau where the gradient of the cost function vanishes exponentially with the number of qubits, and which arises even for constant depth ansatzes. We remark that in principle one can always find a minimizing direction when training $C_G'$, although this would require a number of shots that increases exponentially with $n_B$. Moreover,

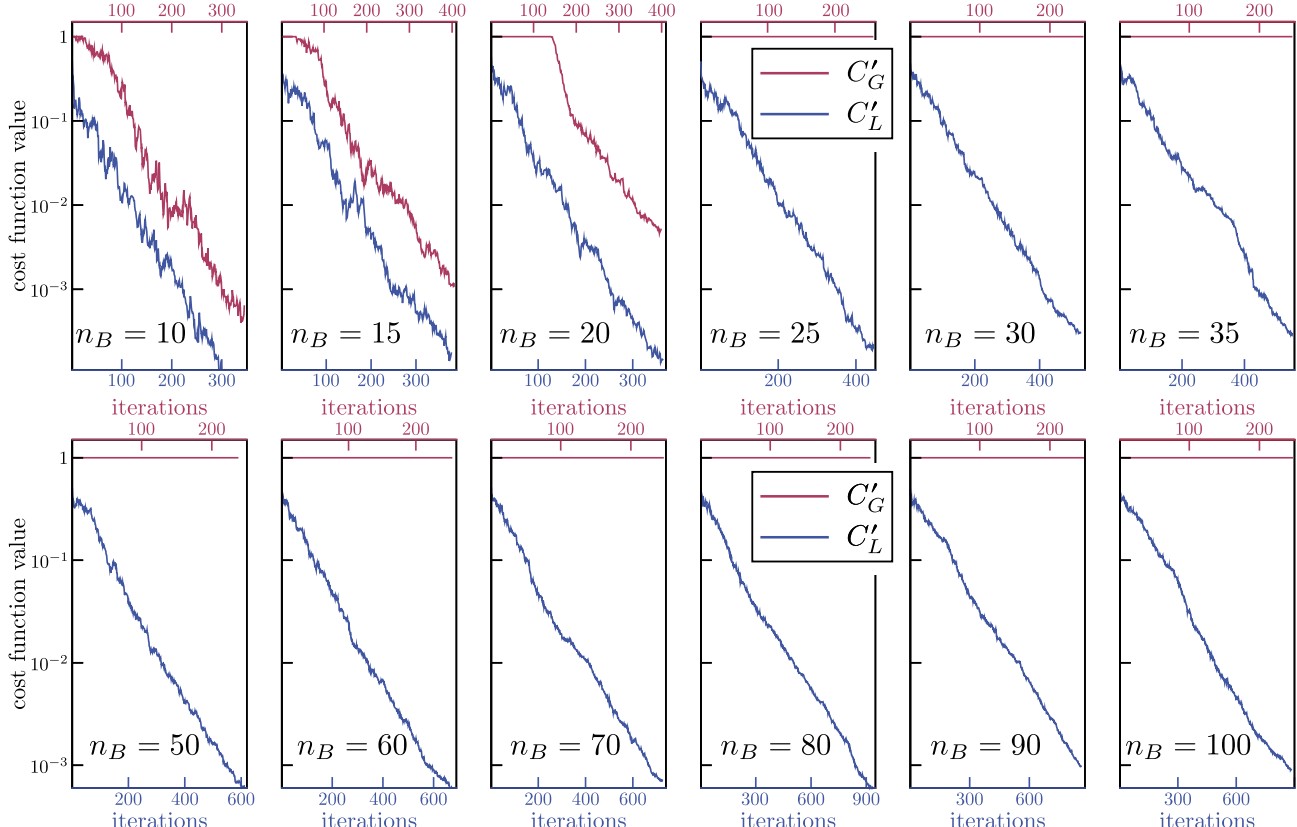

**Fig. 5 Cost versus number of iterations for the quantum autoencoder problem defined by Eqs. (25)–(26).** In all cases we employed two layers of the ansatz shown in Fig. 4, and we set $n_A = 1$, while increasing $n_B = 10, 15, ..., 100$. The top (bottom) axis corresponds to the global cost function $C'_G$ of Eq. (22) (local cost function $C'_L$ of (23)). As can be seen, $C'_G$ can be trained up to $n_B = 20$ qubits, while $C'_L$ trained in all cases. These results indicate that global cost function presents a barren plateau even for a shallow depth ansatz, and this can be avoided by employing a local cost function.

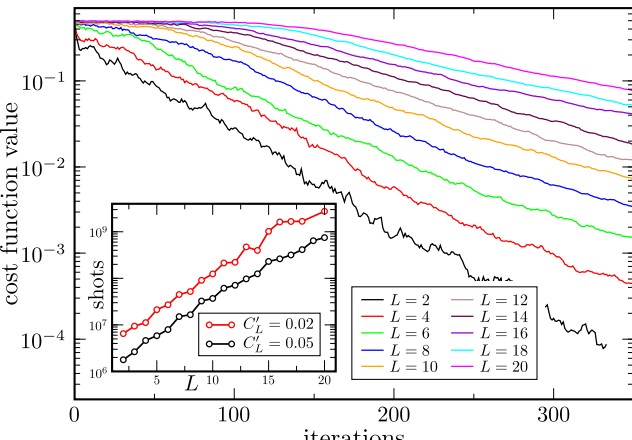

**Fig. 6 Local cost $C'_L$ versus number of iterations for the quantum autoencoder problem in Eqs. (25–26) with $n_B = 10$.** Each curve corresponds to a different number of layers $L$ in the ansatz of Fig. 4 with $L = 2,..., 20$. Curves were averaged over 9 instances of the autoencoder. As the number of layers increases, the optimization becomes harder. Inset: Number of shots needed to reach cost values of $C'_L = 0.02$ and $C'_L = 0.05$ versus number of layers $L$. As $L$ increases the number of shots needed to reach the indicated cost values appears to increase exponentially.

one can see in Fig. 5 that randomly initializing the parameters always leads to $C'_G \approx 1$ due to the narrow gorge phenomenon (see Proposition 1), i.e., where the probability of being near the global minimum vanishes exponentially with $n_B$.

On the other hand, Fig. 5 shows that the barren plateau is avoided when employing a local cost function since we can train $C'_L$ for all considered values of $n_B$. Moreover, as seen in Fig. 5, $C'_L$ can be trained with a small number of shots per cost-function evaluation (as small as 10 shots per evaluation).

Analysis of the $L$-dependence. The power of Theorem 2 is that it gives the scaling in terms of $L$. While one can substitute a function of $n$ for $L$ as we did in Corollary 2, one can also directly study the scaling with $L$ (for fixed $n$). Figure 6 shows the dependence on $L$ when training $C'_L$ for the autoencoder example with $n_A = 1$ and $n_B = 10$. As one can see, the training becomes more difficult as $L$ increases. Specifically, as shown in the inset it appears to become exponentially more difficult, as the number of shots needed to achieve a fixed cost value grows exponentially with $L$. This is consistent with (and hence verifies) our bound on the variance in Theorem 2, which vanishes exponentially in $L$, although we remark that this behavior can saturate for very large $L$[9].

In summary, even though the ansatz employed in our heuristics is beyond the scope of our theorems, we still find cost-function-dependent barren plateaus, indicating that the cost-function dependent barren plateau phenomenon might be more general and go beyond our analytical results.

## Discussion

While scaling results have been obtained for classical neural networks[45], very few such results exist for the trainability of parametrized quantum circuits, and more generally for quantum neural networks. Hence, rigorous scaling results are urgently

needed for VQAs, which many researchers believe will provide the path to quantum advantage with near-term quantum computers. One of the few such results is the barren plateau theorem of ref. [9], which holds for VQAs with deep, hardware-efficient ansatzes.

In this work, we proved that the barren plateau phenomenon extends to VQAs with randomly initialized shallow Alternating Layered Ansatzes. The key to extending this phenomenon to shallow circuits was to consider the locality of the operator $O$ that defines the cost function $C$. Theorem 1 presented a universal upper bound on the variance of the gradient for global cost functions, i.e., when $O$ is a global operator. Corollary 1 stated the asymptotic scaling of this upper bound for shallow ansatzes as being exponentially decaying in $n$, indicating a barren plateau. Conversely, Theorem 2 presented a universal lower bound on the variance of the gradient for local cost functions, i.e., when $O$ is a sum of local operators. Corollary 2 notes that for shallow ansatzes this lower bound decays polynomially in $n$. Taken together, these two results show that barren plateaus are cost-function-dependent, and they establish a connection between locality and trainability.

In the context of chemistry or materials science, our present work can inform researchers about which transformation to use when mapping a fermionic Hamiltonian to a spin Hamiltonian[46], i.e., Jordan–Wigner versus Bravyi–Kitaev[47]. Namely, the Bravyi–Kitaev transformation often leads to more local Pauli terms, and hence (from Corollary 2) to a more trainable cost function. This fact was recently numerically confirmed[48].

Moreover, the fact that Corollary 2 is valid for arbitrary input quantum states may be useful when constructing variational ansatzes. For example, one could propose a growing ansatz method where one appends $\log(n)$ layers of the hardware-efficient ansatz to a previously trained (hence fixed) circuit. This could then lead to a layer-by-layer training strategy where the previously trained circuit can correspond to multiple layers of the same hardware-efficient ansatz.

We remark that our definition of a global operator (local operator) is one that is both non-local (local) and many body (few body). Therefore, the barren plateau phenomenon could be due to the many-bodiness of the operator rather than the non-locality of the operator; we leave the resolution of this question to future work. On the other hand, our Theorem 1 rules out the possibility that barren plateaus could be due to cardinality, i.e., the number of terms in $O$ when decomposed as a sum of Pauli products[49]. Namely, case (ii) of this theorem implies barren plateaus for $O$ of essentially arbitrary cardinality, and hence cardinality is not the key variable at work here.

We illustrated these ideas for two examples VQAs. In Fig. 2, we considered a simple state-preparation example, which allowed us to delve deeper into the cost landscape and uncover another phenomenon that we called a narrow gorge, stated precisely in Proposition 1. In Fig. 5, we studied the more important example of quantum autoencoders, which have generated significant interest in the quantum machine learning community. Our numerics showed the effects of barren plateaus: for more than 20 qubits we were unable to minimize the global cost function introduced in[11]. To address this, we introduced a local cost function for quantum autoencoders, which we were able to minimize for system sizes of up to 100 qubits.

There are several directions in which our results could be generalized in future work. Naturally, we hope to extend the narrow gorge phenomenon in Proposition 1 to more general VQAs. In addition, we hope in the future to unify our theorems 1 and 2 into a single result that bounds the variance as a function of a parameter that quantifies the locality of $O$. This would further solidify the connection between locality and trainability.

Moreover, our numerics suggest that our theorems (which are stated for exact 2-designs) might be extendable in some form to ansatzes composed of simpler blocks, like approximate 2-designs[39].

We emphasize that while our theorems are stated for a hardware-efficient ansatz and for costs that are of the form (6), it remains an interesting open question as to whether other ansatzes, cost function, and architectures exhibit similar scaling behavior as that stated in our theorems. For instance, we have recently shown[50] that our results can be extended to a more general type of Quantum Neural Network called dissipative quantum neural networks[51]. Another potential example of interest could be the unitary-coupled cluster (UCC) ansatz in chemistry[52], which is intended for use in the $\mathcal{O}(\text{poly}(n))$ depth regime[34]. Therefore it is important to study the key mathematical features of an ansatz that might allow one to go from trainability for $\mathcal{O}(\log n)$ depth (which we guarantee here for local cost functions) to trainability for $\mathcal{O}(\text{poly } n)$ depth.

Finally, we remark that some strategies have been developed to mitigate the effects of barren plateaus[32,33,53,54]. While these methods are promising and have been shown to work in certain cases, they are still heuristic methods with no provable guarantees that they can work in generic scenarios. Hence, we believe that more work needs to be done to better understand how to prevent, avoid, or mitigate the effects of barren plateaus.

## Methods

In this section, we provide additional details for the results in the main text, as well as a sketch of the proofs for our main theorems. We note that the proof of Theorem 2 comes before that of Theorem 1 since the latter builds on the former. More detailed proofs of our theorems are given in the Supplementary Information.

**Variance of the cost function partial derivative.** Let us first discuss the formulas we employed to compute $\text{Var}[\partial_\nu C]$. Let us first note that without loss of generality, any block $W_{kl}(\boldsymbol{\theta}_{kl})$ in the Alternating Layered Ansatz can be written as a product of $\zeta_{kl}$ independent gates from a gate alphabet $\mathcal{A} = \{G_\mu(\theta)\}$ as

$$W_{kl}(\boldsymbol{\theta}_{kl}) = G_{\zeta_{kl}}(\theta_{kl}^{\zeta_{kl}})\dots G_\nu(\theta_{kl}^\nu)\dots G_1(\theta_{kl}^1) , \quad (27)$$

where each $\theta_{kl}^\nu$ is a continuous parameter. Here, $G_\nu(\theta_{kl}^\nu) = R_\nu(\theta_{kl}^\nu)Q_\nu$ where $Q_\nu$ is an unparametrized gate and $R_\nu(\theta_{kl}^\nu) = e^{-i\theta_{kl}^\nu \sigma_\nu/2}$ with $\sigma_\nu$ a Pauli operator. Note that $W_{kL}$ denotes a block in the last layer of $V(\boldsymbol{\theta})$.

For the proofs of our results, it is helpful to conceptually break up the ansatz as follows. Consider a block $W_{kl}(\boldsymbol{\theta}_{kl})$ in the $l$th layer of the ansatz. For simplicity, we henceforth use $W$ to refer to a given $W_{kl}(\boldsymbol{\theta}_{kl})$. Let $S_w$ denote the $m$-qubit subsystem that contains the qubits $W$ acts on, and let $S_{\overline{w}}$ be the $(n - m)$ subsystem on which $W$ acts trivially. Similarly, let $\mathcal{H}_w$ and $\mathcal{H}_{\overline{w}}$ denote the Hilbert spaces associated with $S_w$ and $S_{\overline{w}}$, respectively. Then, as shown in Fig. 3a, $V(\boldsymbol{\theta})$ can be expressed as

$$V(\boldsymbol{\theta}) = V_{\text{R}}(\mathbb{1}_{\overline{w}} \otimes W)V_{\text{L}} . \quad (28)$$

Here, $\mathbb{1}_{\overline{w}}$ is the identity on $\mathcal{H}_{\overline{w}}$, and $V_{\text{R}}$ contains the gates in the (forward) light-cone $\mathcal{L}$ of $W$, i.e., all gates with at least one input qubit causally connected to the output qubits of $W$. The latter allows us to define $S_\mathcal{L}$ as the subsystem of all qubits in $\mathcal{L}$.

Let us here recall that the Alternating Layered Ansatz can be implemented with either a 1D or 2D square connectivity as schematically depicted in Fig. 3c. We remark that the following results are valid for both cases as the light-cone structure will be the same. Moreover, the notation employed in our proofs applies to both the 1D and 2D cases. Hence, there is no need to refer to the connectivity dimension in what follows.

Let us now assume that $\theta^\nu$ is a parameter inside a given block $W$, we obtain from (6), (27), and (28)

$$\partial_\nu C = \frac{i}{2}\text{Tr}\Big[(\mathbb{1}_{\overline{w}} \otimes W_{\text{B}})V_{\text{L}}\rho V_{\text{L}}^\dagger(\mathbb{1}_{\overline{w}} \otimes W_{\text{B}}^\dagger) \\ \times [\mathbb{1}_{\overline{w}} \otimes \sigma_\nu, (\mathbb{1}_{\overline{w}} \otimes W_{\text{A}}^\dagger)V_{\text{R}}^\dagger O V_{\text{R}}(\mathbb{1}_{\overline{w}} \otimes W_{\text{A}})]\Big] , \quad (29)$$

with

$$W_{\text{B}} = \prod_{\mu=1}^{\nu-1} G_\mu(\theta^\mu) , \quad \text{and} \quad W_{\text{A}} = \prod_{\mu=\nu}^{\zeta} G_\mu(\theta^\mu) . \quad (30)$$

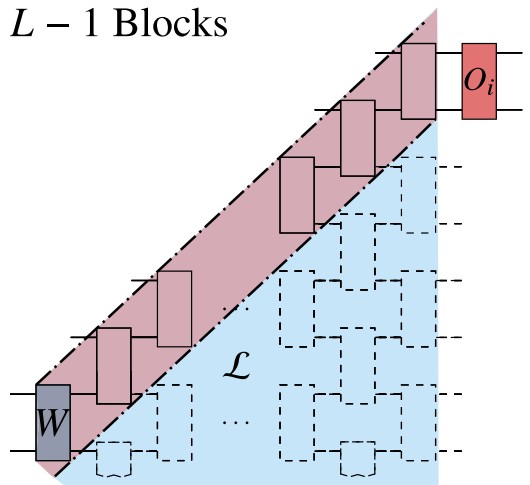

**Fig. 7 The block $W$ is in the first layer of $V(\boldsymbol{\theta})$, and the operator $O_i$ acts on the topmost $m$ qubits in the forward light-cone $\mathcal{L}$ of $W$.** Dashed thick lines indicate the backward light-cone of $O_i$. All but $L-1$ blocks simplify to identity in $\Omega_{\boldsymbol{qp}}$ of Eq. (34).

Finally, from (29) we can derive a general formula for the variance:

$$\mathrm{Var}[\partial_\nu C] = \frac{2^{m-1}\mathrm{Tr}[\sigma_\nu^2]}{(2^{2m}-1)^2} \sum_{\substack{\boldsymbol{pq} \\ \boldsymbol{p'q'}}} \langle \Delta\Omega_{\boldsymbol{qp}}^{\boldsymbol{q'p'}} \rangle_{V_R} \langle \Delta\Psi_{\boldsymbol{pq}}^{\boldsymbol{p'q'}} \rangle_{V_L} \quad (31)$$

which holds if $W_A$ and $W_B$ form independent 2-designs. Here, the summation runs over all bitstrings $\boldsymbol{p}, \boldsymbol{q}, \boldsymbol{p'}, \boldsymbol{q'}$ of length $2^{n-m}$. In addition, we defined

$$\Delta\Omega_{\boldsymbol{qp}}^{\boldsymbol{q'p'}} = \mathrm{Tr}[\Omega_{\boldsymbol{qp}}\Omega_{\boldsymbol{q'p'}}] - \frac{\mathrm{Tr}[\Omega_{\boldsymbol{qp}}]\mathrm{Tr}[\Omega_{\boldsymbol{q'p'}}]}{2^m}, \quad (32)$$

$$\Delta\Psi_{\boldsymbol{pq}}^{\boldsymbol{p'q'}} = \mathrm{Tr}[\Psi_{\boldsymbol{pq}}\Psi_{\boldsymbol{p'q'}}] - \frac{\mathrm{Tr}[\Psi_{\boldsymbol{pq}}]\mathrm{Tr}[\Psi_{\boldsymbol{p'q'}}]}{2^m}, \quad (33)$$

where $\mathrm{Tr}_{\bar{w}}$ indicates the trace over subsystem $S_{\bar{w}}$, and $\Omega_{\boldsymbol{qp}}$ and $\Psi_{\boldsymbol{qp}}$ are operators on $\mathcal{H}_w$ defined as

$$\Omega_{\boldsymbol{qp}} = \mathrm{Tr}_{\bar{w}}\left[(|\boldsymbol{p}\rangle\langle\boldsymbol{q}| \otimes \mathbb{1}_w)V_R^\dagger O V_R\right], \quad (34)$$

$$\Psi_{\boldsymbol{pq}} = \mathrm{Tr}_{\bar{w}}\left[(|\boldsymbol{q}\rangle\langle\boldsymbol{p}| \otimes \mathbb{1}_w)V_L \rho V_L^\dagger\right]. \quad (35)$$

We derive Eq. (31) in the Supplementary Note 4.

**Computing averages over $V$.** Here we introduce the main tools employed to compute quantities of the form $\langle\ldots\rangle_V$. These tools are used throughout the proofs of our main results.

Let us first remark that if the blocks in $V(\boldsymbol{\theta})$ are independent, then any average over $V$ can be computed by averaging over the individual blocks, i.e., $\langle\ldots\rangle_V = \langle\ldots\rangle_{W_{11},\ldots,W_{kl},\ldots} = \langle\ldots\rangle_{V_L,W,V_R}$. For simplicity let us first consider the expectation value over a single block $W$ in the ansatz. In principle $\langle\ldots\rangle_W$ can be approximated by varying the parameters in $W$ and sampling over the resulting $2^m \times 2^m$ unitaries. However, if $W$ forms a $t$-design, this procedure can be simplified as it is known that sampling over its distribution yields the same properties as sampling random unitaries from the unitary group with respect to the unique normalized Haar measure.

Explicitly, the Haar measure is a uniquely defined left and right-invariant measure over the unitary group $d\mu(W)$, such that for any unitary matrix $A \in U(2^m)$ and for any function $f(W)$ we have

$$\int_{U(2^m)} d\mu(W)f(W) = \int d\mu(W)f(AW)$$
$$= \int d\mu(W)f(WA), \quad (36)$$

where the integration domain is assumed to be $U(2^m)$ throughout this work. Consider a finite set $\{W_y\}_{y\in Y}$ (of size $|Y|$) of unitaries $W_y$, and let $P_{(t,t)}(W)$ be an arbitrary polynomial of degree at most $t$ in the matrix elements of $W$ and at most $t$

in those of $W^\dagger$. Then, this finite set is a $t$-design if[38]

$$\langle P_{(t,t)}(W)\rangle_w = \frac{1}{|Y|} \cdot \sum_{y\in Y} P_{(t,t)}(W_y)$$
$$= \int d\mu(W)P_{(t,t)}(W). \quad (37)$$

From the general form of $C$ in Eq. (6) we can see the cost function is a polynomial of degree at most 2 in the matrix elements of each block $W_{kl}$ in $V(\boldsymbol{\theta})$, and at most 2 in those of $(W_{kl})^\dagger$. Then, if a given block $W$ forms a 2-design, one can employ the following elementwise formula of the Weingarten calculus[55,56] to explicitly evaluate averages over $W$ up to the second moment:

$$\int d\mu(W)w_{ij}w_{i'j'}^* = \frac{\delta_{ii'}\delta_{jj'}}{2^m}, \int d\mu(W)w_{i_1j_1}w_{i_2j_2}w_{i'_1j'_1}^*w_{i'_2j'_2}^* = \frac{1}{2^{2m}-1}\left(\Delta_1 - \frac{\Delta_2}{2^m}\right) \quad (38)$$

where $w_{ij}$ are the matrix elements of $W$, and

$$\Delta_1 = \delta_{i_1i'_1}\delta_{i_2i'_2}\delta_{j_1j'_1}\delta_{j_2j'_2} + \delta_{i_1i'_2}\delta_{i_2i'_1}\delta_{j_1j'_2}\delta_{j_2j'_1},$$
$$\Delta_2 = \delta_{i_1i'_1}\delta_{i_2i'_2}\delta_{j_1j'_2}\delta_{j_2j'_1} + \delta_{i_1i'_2}\delta_{i_2i'_1}\delta_{j_1j'_1}\delta_{j_2j'_2}. \quad (39)$$

**Intuition behind the main results.** The goal of this section is to provide some intuition for our main results. Specifically, we show here how the scaling of the cost function variance can be related to the number of blocks we have to integrate to compute $\langle\cdots\rangle_{V_R,V_L}$, the locality of the cost functions, and with the number of layers in the ansatz.

First, we recall from Eq. (38) that integrating over a block leads to a coefficient of the order $1/2^{2m}$. Hence, we see that the more blocks one integrates over, the worse the scaling can be.

We now generalize the warm-up example. Let $V(\boldsymbol{\theta})$ be a single layer of the alternating ansatz of Fig. 3, i.e., $V(\boldsymbol{\theta})$ is a tensor product of $m$-qubit blocks $W_k := W_{k1}$, with $k = 1, \ldots, \xi$ (and with $\xi = n/m$), so that $\theta^\nu$ is in the block $W_{k'}$. In the Supplementary Note 5 we generalize this scenario to the when the input state is not $|\boldsymbol{0}\rangle$, but instead an arbitrary state $\rho$.

From (31), the partial derivative of the global cost function in (1) can be expressed as

$$\mathrm{Var}[\partial_\nu C_G] = \upsilon \prod_{k\neq k'} \left\langle \mathrm{Tr}\left[|0\rangle\langle0|^{\otimes m}W_k|0\rangle\langle0|^{\otimes m}W_k^\dagger\right]^2 \right\rangle_{W_k} \quad (40)$$

where $\upsilon = \frac{(2^m-1)^2\mathrm{Tr}[\sigma_\nu^2]}{2^{2m}(2^{m+1}-1)^2}$. From (40) we have that in order to compute (40) one needs to integrate over $\xi - 1$ blocks. Then, since each integration leads to a coefficient $1/2^{2m}$ the variance will scale as $\mathcal{O}(1/(2^{2m})^{\xi-1} = \mathcal{O}(1/2^{2n})$. Hence, the scaling of the variance gets worse for each block we integrate (such that the block acts on qubits we are measuring).

On the other hand, for a local cost let us consider a single term in (3) where $j \in S_{\bar{k}}$, so that

$$\mathrm{Var}[\partial_\nu C_L] \propto \frac{\upsilon}{n^2} \left\langle \mathrm{Tr}\left[(|0\rangle\langle0|_j \otimes \mathbb{1}_{\bar{j}})W_{\bar{k}}|0\rangle\langle0|^{\otimes m}W_{\bar{k}}^\dagger\right]^2 \right\rangle_{W_{\bar{k}}}. \quad (41)$$

Here, in contrast to the global case, we only have to integrate over a single block irrespective of the total number of qubits. Hence, we now find that the variance scales as $\mathcal{O}(1/n^2)$, where we remark that the scaling is essentially given by the prefactor $1/n^2$ in (3).

Let us now briefly provide some intuition as to why the scaling of local cost gradients becomes exponentially vanishing with the number of layers as in Theorem 2. Consider the case when $V(\boldsymbol{\theta})$ contains $L$ layers of the ansatz in Fig. 3. Moreover, as shown in Fig. 7, let $W$ be in the first layer, and let $O_i$ act on the $m$ topmost qubits of $\mathcal{L}$. As schematically depicted in Fig. 7, we now have to integrating over $L - 1$ blocks. Then, as proved in the Supplementary Note 5, integrating over a block leads to a coefficient $2^{m/2}/(2^m + 1)$. Hence, after integrating $L - 1$ times, we obtain a coefficient $2^{m(L-1)/2}/(2^m + 1)^{L-1}$ which vanishes no faster than $\Omega(1/\mathrm{poly}(n))$ if $mL \in \mathcal{O}(\log(n))$.

As we discuss below, for more general scenarios the computation of $\mathrm{Var}[\partial_\nu C]$ becomes more complex.

**Sketch of the proof of the main theorems.** Here we present a sketch of the proof of Theorems 1 and 2. We refer the reader to the Supplementary Information for a detailed version of the proofs.

As mentioned in the previous subsection, if each block in $V(\boldsymbol{\theta})$ forms a local 2-design, then we can explicitly calculate expectation values $\langle\ldots\rangle_W$ via (38). Hence, to compute $\langle\Delta\Omega_{\boldsymbol{qp}}^{\boldsymbol{p'q'}}\rangle_{V_R}$, and $\langle\Delta\Psi_{\boldsymbol{pq}}^{\boldsymbol{p'q'}}\rangle_{V_L}$ in (31), one needs to algorithmically integrate over each block using the Weingarten calculus. In order to make such computation tractable, we employ the tensor network representation of quantum circuits.

For the sake of clarity, we recall that any two-qubit gate can be expressed as $U = \sum_{ijkl} U_{ijkl}|ij\rangle\langle kl|$, where $U_{ijkl}$ is a $2 \times 2 \times 2 \times 2$ tensor. Similarly, any block in the ansatz can be considered as a $2^{\frac{m}{2}} \times 2^{\frac{m}{2}} \times 2^{\frac{m}{2}} \times 2^{\frac{m}{2}}$ tensor. As schematically shown in Fig. 8a, one can use the circuit description of $\Omega_{\boldsymbol{qp}}^i$ and $\Psi_{\boldsymbol{pq}}$ to derive the tensor

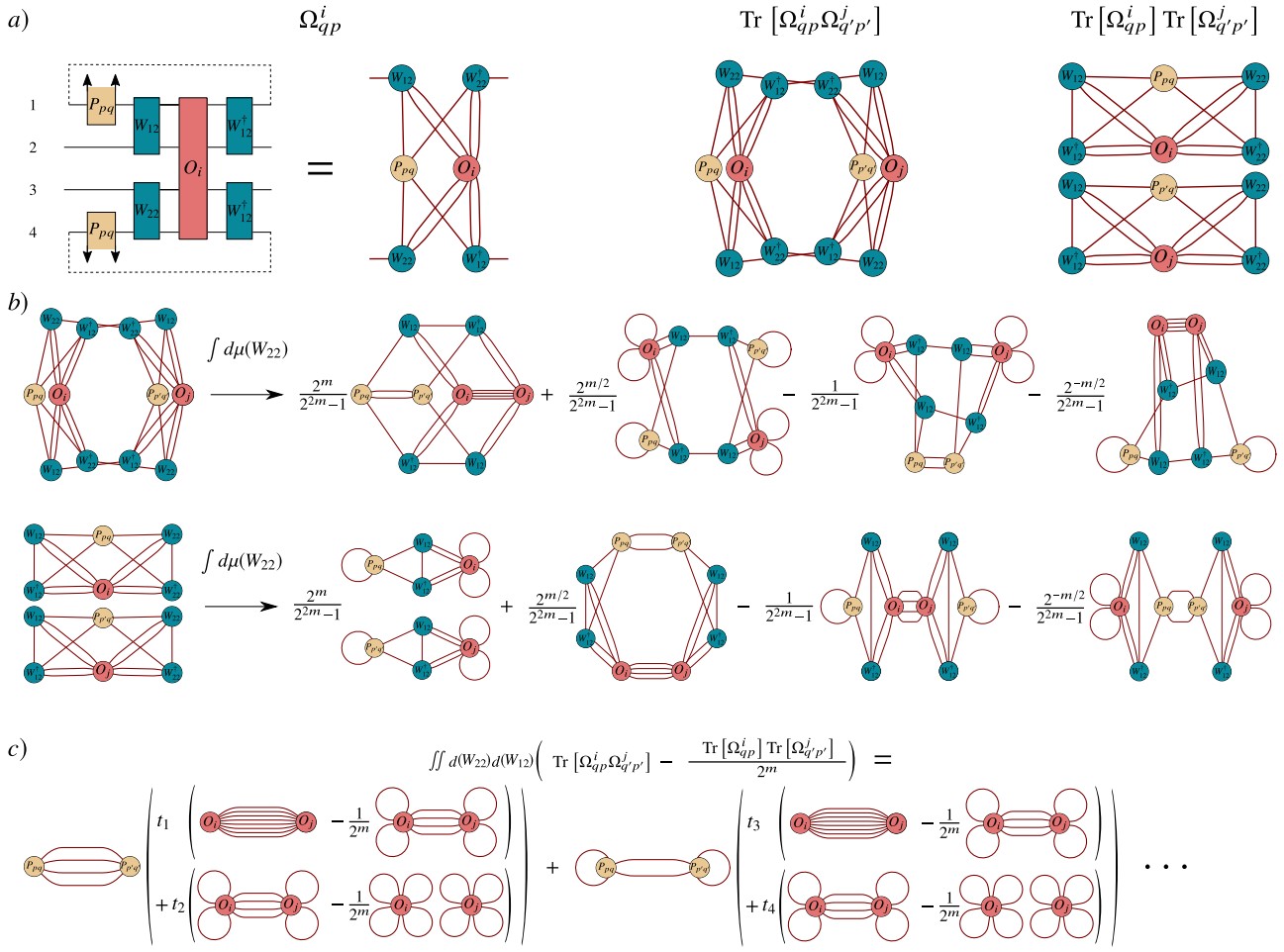

**Fig. 8 Tensor-network representations of the terms relevant to Var[∂ᵥC]. a** Representation of $\Omega_{qp}^i$ of Eq. (34) (left), where the superscript indicates that $O$ is replaced by $O_i$. In this illustration, we show the case of $n = 2m$ qubits, and we denote $P_{pq} = |q\rangle\langle p|$. We also show the representation of $\text{Tr}[\Omega_{qp}^i \Omega_{q'p'}^j]$ (middle) and of $\text{Tr}[\Omega_{qp}^i]\text{Tr}[\Omega_{q'p'}^j]$ (right). **b** By means of the Weingarten calculus we can algorithmically integrate over each block in the ansatz. After each integration one obtains four new tensors according to Eq. (38). Here we show the tensor obtained after the integrations $\int d\mu(W)\text{Tr}[\Omega_{qp}^i \Omega_{q'p'}^j]$ and $\int d\mu(W)\text{Tr}[\Omega_{qp}^i]\text{Tr}[\Omega_{q'p'}^j]$, which are needed to compute $\langle \Delta\Omega_{qp}^{q'p'}\rangle_{V_R}$ as in Eq. (31). **c** As shown in the Supplementary Note 1, the result of the integration is a sum of the form (49), where the deltas over **p**, **q**, **p′**, and **q′** arise from the contractions between $P_{pq}$ and $P_{p'q'}$.

network representation of terms such as $\text{Tr}[\Omega_{qp}^i \Omega_{q'p'}^j]$. Here, $\Omega_{qp}^i$ is obtained from (34) by simply replacing $O$ with $O_i$.

In Fig. 8b we depict an example where we employ the tensor network representation of $\Omega_{qp}^i$ to compute the average of $\text{Tr}[\Omega_{qp}^i \Omega_{q'p'}^j]$, and $\text{Tr}[\Omega_{qp}^i]\text{Tr}[\Omega_{q'p'}^j]$. As expected, after each integration one obtains a sum of four new tensor networks according to Eq. (38).

**Proof of Theorem 2.** Let us first consider an $m$-local cost function $C$ where $O$ is given by (16), and where $\widehat{O}_i$ acts nontrivially in a given subsystem $S_k$ of $\mathcal{S}$. In particular, when $\widehat{O}_i$ is of this form the proof is simplified, although the more general proof is presented in the Supplementary Note 6. If $S_k \not\subset S_{\mathcal{L}}$ we find $\Omega_{qp}^i \propto \mathbb{1}_w$, and hence

$$\text{Tr}[\Omega_{qp}^i \Omega_{q'p'}^j] - \frac{\text{Tr}[\Omega_{qp}^i]\text{Tr}[\Omega_{q'p'}^j]}{2^m} = 0 \ . \tag{42}$$

The latter implies that we only have to consider the operators $\widehat{O}_i$ which act on qubits inside of the forward light-cone $\mathcal{L}$ of $W$.

Then, as shown in the Supplementary Information

$$\left\langle \text{Tr}[\Omega_{qp}^i \Omega_{q'p'}^j] - \frac{\text{Tr}[\Omega_{qp}^i]\text{Tr}[\Omega_{q'p'}^j]}{2^m} \right\rangle_{V_R} \propto \epsilon(\widehat{O}_i) \ . \tag{43}$$

Here we remark that the proportionality factor contains terms of the form $\delta_{(p,q)_{S_{\overline{w}}^+}} \delta_{(p',q')_{S_{\overline{w}}^+}} \delta_{(p,q)_{S_{\overline{w}}^-}} \delta_{(p',q')_{S_{\overline{w}}^-}}$ (where $S_{\overline{w}}^+ \cup S_{\overline{w}}^- = S_{\overline{w}}$), which arises from the different tensor contractions of $P_{pq} = |q\rangle\langle p|$ in Fig. 8c. It is then straightforward to

show that

$$\sum_{\substack{pq \\ p'q'}} \delta_{(p,q)_{S_{\overline{w}}^+}} \delta_{(p',q')_{S_{\overline{w}}^+}} \delta_{(p,q)_{S_{\overline{w}}^-}} \delta_{(p',q')_{S_{\overline{w}}^-}} \left\langle \Delta\Psi_{pq}^{p'q'}\right\rangle_{V_L}$$
$$= \left\langle D_{\text{HS}}\left(\tilde{\rho}^-, \text{Tr}_w[\tilde{\rho}^-] \otimes \frac{\mathbb{1}}{2^m}\right)\right\rangle_{V_L} \ , \tag{44}$$

where we define $\tilde{\rho}^-$ as the reduced states of $\tilde{\rho} = V_L \rho V_L^\dagger$ in the Hilbert spaces associated with subsystems $S_w \cup S_{\overline{w}}^-$. Here we recall that $D_{HS}$ is the Hilbert–Schmidt distance $D_{HS}(\rho, \sigma) = \text{Tr}[(\rho - \sigma)^2]$.

By employing properties of $D_{HS}$ one can show (see Supplementary Note 6)

$$D_{\text{HS}}\left(\tilde{\rho}^-, \text{Tr}_w[\tilde{\rho}^-] \otimes \frac{\mathbb{1}}{2^m}\right) \geq \frac{D_{\text{HS}}\left(\tilde{\rho}_w, \frac{\mathbb{1}}{2^m}\right)}{2^{m(L-l+2)/2}} \ , \tag{45}$$

where $\tilde{\rho}_w = \text{Tr}_{S_{\overline{w}}^-}[\tilde{\rho}^-]$. We can then leverage the tensor network representation of quantum circuits to algorithmically integrate over each block in $V_L$ and compute $\langle D_{\text{HS}}(\tilde{\rho}_w, \frac{\mathbb{1}}{2^m})\rangle_{V_L}$. One finds

$$\left\langle D_{\text{HS}}\left(\tilde{\rho}_w, \frac{\mathbb{1}}{2^m}\right)\right\rangle_{V_L} = \sum_{\substack{(k,k')\in k_L_{\text{B}} \\ k' \geq k}} t_{k,k'}\epsilon(\rho_{k,k'}) \ , \tag{46}$$

with $t_{k,k'} \geq \frac{2^{ml}}{(2^m+1)^{2^l}} \forall k, k'$, and $\epsilon(\rho_{k,k'})$ defined in Theorem 2. Combining these results leads to Theorem 2. Moreover, as detailed in the Supplementary information, Theorem 2 is also valid when $O$ is of the form (16).

**Proof of Theorem 1.** Let us now provide a sketch of the proof of Theorem 1, case (i). Here we denote for simplicity $\widehat{O}_k := \widehat{O}_{1k}$. We leave the proof of case (ii) for the

Supplementary Note 7. In this case there are now operators $O_i$ which act outside of the forward light-cone $\mathcal{L}$ of $W$. Hence, it is convenient to include in $V_R$ not only all the gates in $\mathcal{L}$ but also all the blocks in the final layer of $V(\boldsymbol{\theta})$ (i.e., all blocks $W_{kL}$, with $k = 1, \ldots \xi$). We can define $S_{\overline{\mathcal{L}}}$ as the compliment of $S_{\mathcal{L}}$, i.e., as the subsystem of all qubits which are not in $\mathcal{L}$ (with associated Hilbert-space $\mathcal{H}_{\overline{\mathcal{L}}}$). Then, we have $V_R = V_{\mathcal{L}} \otimes V_{\overline{\mathcal{L}}}$ and $|\boldsymbol{q}\rangle\langle\boldsymbol{p}| = |\boldsymbol{q}\rangle\langle\boldsymbol{p}|_{\mathcal{L}} \otimes |\boldsymbol{q}\rangle\langle\boldsymbol{p}|_{\overline{\mathcal{L}}}$, where we define $V_{\overline{\mathcal{L}}} := \bigotimes_{k \in k_{\overline{\mathcal{L}}}} W_{kL}$, $|\boldsymbol{q}\rangle\langle\boldsymbol{p}|_{\mathcal{L}} := \bigotimes_{k \in k_{\mathcal{L}}} |\boldsymbol{q}\rangle\langle\boldsymbol{p}|_k$, and $|\boldsymbol{q}\rangle\langle\boldsymbol{p}|_{\overline{\mathcal{L}}} := \bigotimes_{k' \in k_{\overline{\mathcal{L}}}} |\boldsymbol{q}\rangle\langle\boldsymbol{p}|_{k'}$. Here, we define $k_{\mathcal{L}} := \{k : S_k \subseteq S_{\mathcal{L}}\}$ and $k_{\overline{\mathcal{L}}} := \{k : S_k \subseteq S_{\overline{\mathcal{L}}}\}$, which are the set of indices whose associated qubits are inside and outside $\mathcal{L}$, respectively. We also write $O = c_0 \mathbb{1} + c_1 \hat{O}_{\mathcal{L}} \otimes \hat{O}_{\overline{\mathcal{L}}}$, where we define $\hat{O}_{\mathcal{L}} := \bigotimes_{k \in k_{\mathcal{L}}} \hat{O}_k$ and $\hat{O}_{\overline{\mathcal{L}}} := \bigotimes_{k' \in k_{\overline{\mathcal{L}}}} \hat{O}_{k'}$.

Using the fact that the blocks in $V(\boldsymbol{\theta})$ are independent we can now compute $\langle \Delta \Omega_{\boldsymbol{q}\boldsymbol{p}}^{\boldsymbol{q}'\boldsymbol{p}'} \rangle_{V_R} = \langle \Delta \Omega_{\boldsymbol{q}\boldsymbol{p}}^{\boldsymbol{q}'\boldsymbol{p}'} \rangle_{V_{\overline{\mathcal{L}}}, V_{\mathcal{L}}}$. Then, from the definition of $\Omega_{pq}$ in Eq. (34) and the fact that one can always express

$$\left\langle \Delta \Omega_{\boldsymbol{q}\boldsymbol{p}}^{\boldsymbol{q}'\boldsymbol{p}'} \right\rangle_{V_R} = \left\langle \mathrm{Tr}[\Omega_{\boldsymbol{q}\boldsymbol{p}}^{\mathcal{L}} \Omega_{\boldsymbol{q}'\boldsymbol{p}'}^{\mathcal{L}}] - \frac{\mathrm{Tr}[\Omega_{\boldsymbol{q}\boldsymbol{p}}^{\mathcal{L}}] \mathrm{Tr}[\Omega_{\boldsymbol{q}'\boldsymbol{p}'}^{\mathcal{L}}]}{2^m} \right\rangle_{V_{\mathcal{L}}} \times \left( \prod_{k \in k_{\overline{\mathcal{L}}}} \langle \Omega_k \rangle_{W_{kL}} \right), \quad (47)$$

with

$$\Omega_{\boldsymbol{q}\boldsymbol{p}}^{\mathcal{L}} = \mathrm{Tr}_{\mathcal{L} \cap \overline{w}} \left[ (|\boldsymbol{p}\rangle\langle\boldsymbol{q}| \otimes \mathbb{1}_w) V_{\mathcal{L}} O^{\mathcal{L}} V_{\mathcal{L}} \right]$$

$$\Omega_k = \mathrm{Tr}\left[ |\boldsymbol{p}\rangle\langle\boldsymbol{q}|_k W_{kL}^{\dagger} \hat{O}_k W_{kL} \right] \mathrm{Tr}\left[ |\boldsymbol{p}'\rangle\langle\boldsymbol{q}'|_k W_{kL}^{\dagger} \hat{O}_k W_{kL} \right]$$

and where $\mathrm{Tr}_{\mathcal{L} \cap \overline{w}}$ indicates the partial trace over the Hilbert-space associated with the qubits in $S_{\mathcal{L}} \cap S_{\overline{w}}$. As detailed in the Supplementary Information we can use Eq. (38) to show that

$$\langle \Omega_k \rangle_{W_{kL}} \leqslant \frac{r_k^2 \left( \delta_{(\boldsymbol{p},\boldsymbol{q})_{S_k}} \delta_{(\boldsymbol{p}',\boldsymbol{q}')_{S_k}} + \delta_{(\boldsymbol{p},\boldsymbol{q}')_{S_k}} \delta_{(\boldsymbol{p}',\boldsymbol{q})_{S_k}} \right)}{2^{2m} - 1}. \quad (48)$$

On the other hand, as shown in the Supplementary Note 7 (and as schematically depicted in Fig. 8c), when computing the expectation value $\langle \ldots \rangle_{V_{\mathcal{L}}}$ in (47), one obtains

$$\left\langle \mathrm{Tr}[\Omega_{\boldsymbol{q}\boldsymbol{p}}^{\mathcal{L}} \Omega_{\boldsymbol{q}'\boldsymbol{p}'}^{\mathcal{L}}] - \frac{\mathrm{Tr}[\Omega_{\boldsymbol{q}\boldsymbol{p}}^{\mathcal{L}}] \mathrm{Tr}[\Omega_{\boldsymbol{q}'\boldsymbol{p}'}^{\mathcal{L}}]}{2^m} \right\rangle_{V_{\mathcal{L}}} = \sum_{\tau} t_{\tau}^{ij} \Delta O_{\tau}^{\mathcal{L}} \delta_{\tau}, \quad (49)$$

where we defined $\delta_{\tau} = \delta_{(\boldsymbol{p},\boldsymbol{q})_{S_{\overline{\tau}}}} \delta_{(\boldsymbol{p}',\boldsymbol{q}')_{S_{\overline{\tau}}}} \delta_{(\boldsymbol{p},\boldsymbol{q}')_{S_{\tau}}} \delta_{(\boldsymbol{p}',\boldsymbol{q})_{S_{\tau}}}$, $t_{\tau} \in \mathbb{R}$, $S_{\tau} \cup S_{\overline{\tau}} = S_{\mathcal{L}} \cap S_{\overline{w}}$ (with $S_{\tau} \neq \emptyset$), and

$$\Delta O_{\tau}^{\mathcal{L}} = \mathrm{Tr}_{x_{\tau} y_{\tau}} \left[ \mathrm{Tr}_{z_{\tau}}[O_i] \mathrm{Tr}_{z_{\tau}}[O_j] \right] - \frac{\mathrm{Tr}_{x_{\tau}} \left[ \mathrm{Tr}_{y_{\tau} z_{\tau}}[O_i] \mathrm{Tr}_{y_{\tau} z_{\tau}}[O_j] \right]}{2^m}. \quad (50)$$

Here we use the notation $\mathrm{Tr}_{x_{\tau}}$ to indicate the trace over the Hilbert space associated with subsystem $S_{x_{\tau}}$, such that $S_{x_{\tau}} \cup S_{y_{\tau}} \cup S_{z_{\tau}} = S_{\mathcal{L}}$. As shown in the Supplementary Note 7, combining the deltas in Eqs. (48), and (49) with $\left\langle \Delta \Psi_{\boldsymbol{p}\boldsymbol{q}}^{\boldsymbol{p}'\boldsymbol{q}'} \right\rangle_{V_L}$ leads to Hilbert–Schmidt distances between two quantum states as in (44). One can then use the following bounds $D_{HS}(\rho_1, \rho_2) \leq 2$, $\Delta O_{\tau}^{\mathcal{L}} \leq \prod_{k \in k_{\mathcal{L}}} r_k^2$, and $\sum_{\tau} t_{\tau} \leq 2$, along with some additional simple algebra explained in the Supplementary Information to obtain the upper bound in Theorem 1.

**Ansatz and optimization method.** Here we describe the gradient-free optimization method used in our heuristics. First, we note that all the parameters in the ansatz are randomly initialized. Then, at each iteration, one solves the following sub-space search problem: $\min_{\boldsymbol{s} \in \mathbb{R}^d} C(\boldsymbol{\theta} + A \cdot \boldsymbol{s})$, where $A$ is a randomly generated isometry, and $\boldsymbol{s} = (s_1, \ldots, s_d)$ is a vector of coefficients to be optimized over. We used $d = 10$ in our simulations. Moreover, the training algorithm gradually increases the number of shots per cost-function evaluation. Initially, $C$ is evaluated with 10 shots, and once the optimization reaches a plateau, the number of shots is increased by a factor of $3/2$. This process is repeated until a termination condition on the value of $C$ is achieved, or until we reach the maximum value of $10^5$ shots per function evaluation. While this is a simple variable-shot approach, we remark that a more advanced variable-shot optimizer can be found in ref. [57].

Finally, let us remark that while we employ a sub-space search algorithm, in the presence of barren plateaus all optimization methods will (on average) fail unless the algorithm has a precision (i.e., number of shots) that grows exponentially with $n$. The latter is due to the fact that an exponentially vanishing gradient implies that on average the cost function landscape will essentially be flat, with the slope of the order of $\mathcal{O}(1/2^n)$. Hence, unless one has a precision that can detect such small changes in the cost value, one will not be able to determine a cost minimization direction with gradient-based, or even with black-box optimizers such as the Nelder–Mead method[58–61].

## Data availability
Data generated and analyzed during the current study are available from the corresponding author upon reasonable request.

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

## Acknowledgements
We thank Jacob Biamonte, Elizabeth Crosson, Burak Sahinoglu, Rolando Somma, Guillaume Verdon, and Kunal Sharma for helpful conversations. All authors were supported by the Laboratory Directed Research and Development (LDRD) program of Los Alamos National Laboratory (LANL) under project numbers 20180628ECR (for M.C.), 20190065DR (for A.S., L.C., and P.J.C.), and 20200677PRD1 (for T.V.). M.C. and A.S. were also supported by the Center for Nonlinear Studies at LANL. P.J.C. acknowledges initial support from the LANL ASC Beyond Moore's Law project. This work was also supported by the U.S. Department of Energy (DOE), Office of Science, Office of Advanced Scientific Computing Research, under the Quantum Computing Application Teams program.

## Author contributions
The project was conceived by M.C., L.C., and P.J.C. The manuscript was written by M.C., A.S., T.V., L.C., and P.J.C. T.V. proved Proposition 1. M.C. and A.S. proved Proposition 2 and Theorems 1–2. M.C., A.S., T.V., and P.J.C. proved Corollaries 1–2. M.C., A.S., T.V., L.C., and P.J.C. analyzed the quantum autoencoder. For the numerical results, T.V. performed the simulation in Fig. 2, and L.C. performed the simulation in Fig. 5.

## Competing interests
The authors declare no competing interests.
