## [Peer Review File · Nature Communications]

Reviewers' Comments:

Reviewer #1:

Remarks to the Author:

This article studies the so-called "barren plateaus" phenomenon which occurs in quantum neural networks (QNNs). Roughly, if a sufficiently deep "hardware-efficient" variational ansatz is randomly initialized, then with overwhelming probability the gradient of the cost function is exponentially close to zero and hence training is infeasible. The existence of this phenomenon in the setting of deep circuits follows from well-known concentration of measure results, and was explicitly derived in [1].

While [1] considers the regime of deep circuits where concentration of measure results are applicable, the present article considers the setting of shallow circuits (i.e. constant, log, or polylog depth). Two main results are given. First, if the observable generating the cost function is defined non-locally, then even shallow circuits have the vanishing gradient problem. On the other hand, for a natural family of 1D local cost functions and for hardware-efficient ansatzes of depth $O(\log n)$, the authors obtain a lower bound on the variance of the gradient -- in this case the gradient vanishes polynomially in the system size rather than exponentially. The formal setting considered by the authors will generally not exactly correspond to settings of practical applications (since simplifications are necessarily to obtain rigorous results), but I do not consider this to be a significant downside. The authors also perform a numerical experiment to demonstrate how choosing a local rather than global cost function for training a quantum autoencoder can lead to a more favorable optimization landscape.

Overall, my opinion on this article is that it is a "weak accept". For one, the authors study a regime that is significantly more non-trivial than that of [1], which also appeared in Nature Comm. This is because in the setting of shallow circuits, random circuits no longer form 2-designs and hence we cannot immediately rely on intuition from concentration of measure results in guiding our expectations. In addition to being technically and conceptually nontrivial, it is also certainly very timely as "NISQ devices" come online. In some sense, there is already a great deal of skepticism surrounding the "hardware efficient ansatz" which this paper studies, so from a practical standpoint these results may likely not be especially important in influencing the field. However, as the authors correctly point out, there is a dearth of theoretical results for hybrid quantum-classical algorithms. It's here that I see the most value of this article: it's one of the few non-trivial rigorous results we have on NISQ algorithms, and could be helpful as a starting point for future theoretical work.

So, summarizing my "weak accept" opinion on this article:

Pros:

-- Conceptually and technically, the results are certainly more nontrivial than that of [1] which also appeared in this journal.

-- It's a rare instance of a rigorous theoretical result for NISQ algorithms, and is even applicable to some specific proposals appearing in the literature. Given the lack of theory for NISQ algorithms, any such result is valuable.

-- It gives theoretical evidence in favor of "local" cost functions over "global" cost functions, which could be influential in proposals for some variational algorithms going forward.

Cons:

-- Despite the last bullet point in the "pros" column, I'm somewhat doubtful that these results will substantially influence the field. This is because this article largely gives negative results for the "hardware-efficient ansatz", which as far as I can tell most of the community has already become quite skeptical of. Personally, my main takeaway from this work is that this ansatz is even worse

than many expected, in the sense that it already encounters the vanishing gradient problem at shallow depths for many reasonable choices of cost function. So, this paper gives nice theoretical results about how this ansatz is often bad, but many already considered it bad to begin with.

-- I wasn't quite sure what to make of the "positive" result of Theorem 2/Corollary 2, which says that for shallow variational circuits and for a broad class of local, 1D cost functions, the gradient vanishes not exponentially with system size but rather as $\Omega(1/\text{poly}(n))$. This result feels rather weak to me, for a couple of reasons. (1) It seems we still encounter vanishing gradients in this case, in the sense that we'll still have $\text{norm}(\text{gradient}) \rightarrow 0$ with probability $\rightarrow 1$ as the system size is scaled up, albeit with polynomially-fast convergence rather than exponentially-fast. (2) Moreover, this holds only in the rather weak setting of $O(\log n)$ -depth circuits with 1D, spatially-local cost functions. So, while I certainly found this result mathematically interesting, I actually viewed it more as a negative result for variational algorithms, rather than a positive result as the authors billed it as. Maybe I'm not fully appreciating something here -- perhaps the authors could provide a bit more motivation for this result.

-- On a similar topic as the above complaint, the numerical experiment did show a speedup for the "local cost function" case over the "global cost function" case, but the circuit depth for this experiment was only four. It seems plausible that even the "local" case might quickly become untrainable as the depth is increased. It would have been interesting for the authors to also try to understand how trainability scales with depth in the local case for this model.

-- While the paper is well-written, one downside is that the authors do not provide much intuition for many of their results, instead only giving long brute-force calculations. I appreciated the toy model demonstrating the vanishing gradient phenomenon for shallow circuits with a non-local cost function, but many other results did not have such accompanying intuitive explanations.

Overall, my personal feeling is that the 'pros' outweigh the 'cons', and so I recommend acceptance. There are some corrections below, including a potential small bug I noticed in one of the proofs. Unfortunately, I did not have the bandwidth to carefully check all details of all calculations in the Supplemental Information, but the proofs that I did not have time to carefully check made sense to me.

More detailed comments/corrections:

-- In Eq. (8), I think you mean for V_L and V_R to be swapped. This same mistake occurs throughout the document.

-- Typo in p. 4: "bistrings" \rightarrow "bitstrings"

-- In p. 9 and p. 13, you define a unitary t -design to be a finite set, however continuous distributions can also be t -designs. Maybe you could instead word this as something like, 'this finite set is a unitary t -design if...'

-- In p. 14 Eq. 53, I think the 'F' should be a 'D'. Also, $V_y \rightarrow W_y$.

-- Typo in the expression for $A_{\{qp\}}$ in Eq. (58).

-- In p. 18 Lemma 7, O_1 should be O_2 . Also $\text{Tr}_k[O] \rightarrow \text{Tr}_j[O]$.

-- Unless I'm misunderstanding something, I think there may be a (noncritical) bug in your proof of Proposition 1 in Supplementary Note 2. In particular, you say "the radius of the integration ball vanishes" when $n \rightarrow \infty$. But it appears to me that the radius approaches $-\ln(1-\delta) > 0$ in the limit. If I'm correct about this, this proof should be revised.

-- In p. 21 Eq 98, I think you actually mean $V_L \rho V_L^\dagger$

-- In p. 22, when you reference Eq 103 I think you might mean Eq 102.

[1] McClean, Jarrod R., et al. "Barren plateaus in quantum neural network training landscapes." Nature Communications 9.1 (2018): 1-6.

Reviewer #2:

Remarks to the Author:

The authors claim to establish fundamental limits on the "trainability" of "quantum neural networks" (QNNs). For this, the training of a QNN is assumed to be defined as minimizing a cost function with a gradient descent method evaluated by sampling the output of a parametrized quantum circuit. It is also assumed that trainability can be equated with the asymptotic behavior of the statistical variance of the gradient of parametrized quantum circuits initialized with random parameters. As a function of the depth of a parametrized quantum circuit, the first two moments (mean and variance) of the gradient can vanish to zero exponentially fast. This is known in the literature as the barren plateau phenomenon.

The paper can be of interest to researchers in the field since it describes an important constraint to take into account when designing variational quantum algorithms. One must be careful with the interplay of circuit parametrization, parameter initialization and locality of the cost function.

In more details, the major claims of the paper relate to the "trainability" of quantum neural networks as a function of the depth of the parametrized quantum circuits and the locality of the cost function. The authors show that:

Cost functions with global observables induce barren plateaus even for very shallow circuits. Cost functions with local observables have polynomially vanishing gradients for $O(\log n)$ depth circuits.

The main message of the authors is to exercise care when designing cost functions and hardware-efficient ansatz for variational quantum algorithms. Specifically, for claim 1 this implies that global cost functions exhibit the narrow gorge phenomenon and are therefore exponentially difficult to optimize with random initial parameters for all circuit depth. This is formalized in Theorem 1 which states that global cost functions have barren plateaus. Corollary 1 then enunciates that $O(\text{polylog}(n))$ -depth ansatz with global cost function have vanishing gradients. For claim 2 this implies that local cost function with random initial parameters "can be trained" if the depth of the parametrized circuit is logarithmic in the number of qubits. This is stated formally in Theorem 2 which says that a local cost function has a lower bound on the variance of the gradient. The practical implication in Corollary 2 is that less than $O(\text{polylog}(n))$ -depth ansatz "can be trained" for a local cost function.

The practical consequences of the main claims are illustrated through the numerical simulation of a variational quantum autoencoder. Claim 1 is illustrated in the example by showing that a barren plateau appears in the training of a global cost function when the number of qubits is larger than 20. The result is a consequence of randomly initializing parameters. Claim 2 is illustrated in the example by showing that the local cost function, which is faithful to the global one, can be trained on all quantum registers up to 100 qubits.

For clarity and to avoid ambiguity, we suggest to lean away from the term QNN. It is never used beyond the title and the abstract. We suggest replacing QNN with parameterized quantum circuits (PQCs), which in our opinion could be broader since the results extend beyond ML applications. If the authors want to stick to the term QNN they should motivate it more and even define what are the requirements of a circuit to be a QNN, something that apparently there is no consensus in the literature. The QNN term does not add any additional "ML coolness", and on the contrary, it can cause ambiguity, since not all PQCs and/or not all problems with hardware-efficient ansatz are QNNs.

The authors provide a derivation of their asymptotic bounds for which they introduce a graphical calculation method using tensor networks and we agree the derivation of the theorems and corollaries are rigorous. And even though the numerical example clearly illustrates the

phenomenon, the main claim about “trainability” of the parametrized quantum circuits (PQCs) is not well justified. Success in the trainability of PQCs is not equivalent to existence or prominence of barren plateaus. The proofs and the results are binded to the strategy of randomized initialization AND the usage of the gradient as a training method since the assessment is made by calculating their variance. Therefore, no conclusions can be made about the trainability of the PQC, for example, under a black-box solver, or any other type of solver or computational strategy which exploits the structure of the problem. As pointed in the paper, there have been many strategies proposed to make circuits with barren plateaus trainable and to cope with this issue. Some of these being initialization strategies, meta-learning heuristics, layer-by-layer training, and some others that have come in part as motivation from the original paper by McClean et al.. It would be good to see if the claims from the authors hold even with black-box solvers. At least this is something outside the scope of the theorems derived and therefore the manuscript should be revisited throughout to avoid any claims related to “trainability” and their connection to the locality of the cost function. From their main results and theorems, the authors can claim that there is a link with the locality of the cost function to the presence or absence of barren plateaus, but saying that this separates a PQC design as trainable or not trainable could be simply wrong or at least certainly beyond the scope of this work.

As the authors claim that trainability is a function of the locality of the cost function observable, they need to provide a discussion on how to reconcile the conclusion of this paper: <https://www.nature.com/articles/s41467-020-14454-2>, with their conclusions. In this reference, the authors define a QNN with a global cost function and they claim “absence of a barren plateau in the cost function landscape.” They go further and mention just before the discussion section that the cost function is non-local means and in the words of the authors, “this means that Levy’s lemma-type argumentation does not directly apply.”

Reviewer #3:

Remarks to the Author:

In this paper, the authors discuss trainability of quantum neural networks at the initial stage of the training. This kind of problem has been investigated in the field of classical neural network as well, for example in [Pennington et al.]. This or other related papers/conference proceedings can be quoted. With classical or quantum neural networks, we first construct networks randomly and improve the parameters of the networks to reduce the value of a chosen cost function (or loss function). This is whey, random matrix theory plays a role. To make the value of the cost function smaller, we use gradients of the cost function with respect to the parameters that we improve. The authors discuss this matter in terms of choice of const functions. More precisely, their conclusions state that local cost functions are better than global cost functions with Variational Quantum Algorithms (VQAs) because gradients vanish slower with local cost functions for randomly constructed Alternating Layered Ansatz. To this end the authors utilized Weingarten Calculus, but they should quote [Collins et al.], where they provided exact Weingarten formulas for unitary, orthogonal and symplectic groups. The results of this paper should be known widely because QVAs have a lot of applications and their models have counterparts in the real world. However, the mathematical discussions have not convinced me yet. (I might be looking over something.) I shall start with the question that I think are important.

In the proof of theorem 2 of Supplementary Note 6, the authors use special case where O_i and O_j act only on topmost m qubits in $V_{\mathcal{L}}$. In this case, as the authors wrote, only $L-l$ blocks are non-trivial, and special iterative calculations are possible as in SUP. FIG. 1. However, in general we get after the integration over W four different terms for each, like in Lemma 6. This is why, I have two questions about (130) when such a special case does not hold.

I) Integrating over more of W s, how does it affect the coefficient? How the power $L-l$ changes?

II) After such integration over a W , we usually we get terms with negative signs as in Lemma 6. How did the authors handle these negative terms appearing for each integration over a W ?

These questions are important because a lower-bound is derived in (130).

Besides, I am wondering how (140) average of Hilbert-Schmidt norm was calculated. Explanations should be added because usually square root does not get along with integration.

There are several comments on Lemma 6. First, notations are confusing. Apart from the typo: $d = d*d*d*d = d^4$, the authors should state what are x and y , and which are "12 possible cases". The notations like $H_4 \setminus H_k$ after (65) is confusing. The proof is confusing to me too. First, the equation (71) seems strange because the LHS is the average over W s but the RHS still has W s in Ω s. Perhaps, the authors wanted to write another equation. Second, the authors tried to "deduce (71)" in the proof by using the statement in the lemma to be proved. This structure should be avoided.

In the proof of Proposition 1, "the radius of the integral ball vanishes", but it seems to me that the limit of $n((1-\delta)^{1/n})$ is $-\log(1-\delta)$ and it does not vanish.

Moving to Supplementary Note 4, and my question is what if $\nu = 1$. The result holds if W_A and W_B are both 2-design, but $\nu = 1$ implies that W_B is the identity from (87).

I shall make some minor comments in the rest of letter.

Is the order correct in (8)? In addition, V_R should be defined. In (58), there is a typo. There are typos in Lemma 7 (the second O_1 should be O_2 ?) and in Proposition 1 (θ is missing in the definition of A_δ^G). trace is missing in (103). I found several "topomost" instead of "topmost".

Reference

Pennington, Jeffrey, and Yasaman Bahri. "Geometry of neural network loss surfaces via random matrix theory." Proceedings of the 34th International Conference on Machine Learning-Volume 70. JMLR. org, 2017.

Collins, Benoît, and Piotr Śniady. "Integration with respect to the Haar measure on unitary, orthogonal and symplectic group." Communications in Mathematical Physics 264.3 (2006): 773-795.

Dear Editors and Referees,

We sincerely appreciate the helpful comments and suggestions made by the referees. We have revised our manuscript to address these comments. Specifically, we wish to highlight the following major changes and additions to our work. We believe that all these revisions make our manuscript clearer and conceptually stronger.

1. **New Analytical Results:** We have managed to show that our main theorems are valid for 2-dimensional grid-like circuits such as the ones found in IBM's [34] and Google's [35] quantum device. We believe that this will greatly improve the applicability and scope of our theorems as they can be readily applied for architectures as the ones in current NISQ devices. In order to showcase this fact we have modified Fig. 3 to show how the Alternating Layered Ansatz can be implemented with 1D and 2D connectivity. We have appropriately modified our theorems and we have added a paragraph in the Methods section stating that our results and notation apply to both cases. We remark that our extension to 2D connectivity addresses the first referee's concerns about the generality of our results.
2. **New Numerical Results:** We have added new heuristics in Fig 6. Therein we analyze the trainability of an 11 qubit autoencoder implementation as we increase the number of layers in the ansatz. This allows us to show that the lower bound in Corollary 2 holds even when the blocks in the ansatz (of Fig 4) do not form 2-designs.
3. **New Insights:** We added a significant amount of text to show the scope, relevance, and applicability of our results, including the following.
 - We have added current references [7] and [8] to the introduction, both of which were not available at the time of our original submission. These manuscripts, by the Google group, analyze large-scale implementations of near-term algorithms for chemistry [7] and optimization [8] applications. In both their works they show that employing hardware efficient ansatzes leads to smaller errors due to hardware noise. We hence believe the results in our manuscript are more timely than ever, as layered hardware efficient ansatzes are state-of-the-art ansatz for near-term applications.
 - We have added a new paragraph in the introduction arguing that polynomially (exponentially) vanishing gradients have hope (no hope) of achieving quantum advantage. Hence, we now argue that analyzing the trainability of quantum neural networks is paramount for guaranteeing that quantum advantage can be achieved.
 - In the discussion section we argue that our results can help researchers develop near-term algorithms with trainability guarantees. For example, in the context of chemistry or materials science, our present work can inform researchers about which transformation to use when mapping a fermionic Hamiltonian to a spin Hamiltonian, i.e., Jordan-Wigner versus Bravyi-Kitaev. Namely, the Bravyi-Kitaev transformation often leads to more local Pauli terms, and hence (from Corollary 2) to a more trainable cost function. This fact was recently numerically confirmed [46], where they cite our manuscript.
 - In the discussion section we now highlight the implications of Corollary 2 for developing novel ansatzes and optimization methods. Particularly we propose a layer-by-layer optimization method which is based on the trainability guarantee of Corollary 2 for $\log(n)$ layers of the Alternating Layered Ansatz.
 - We have now added Ref [48], where we analyze the trainability of a different type of Quantum Neural Network (QNN) called, Dissipative QNN (DQNN). The DQNN architecture was recently proposed by K. Beer, et al, and their work was recently published in Nature Communications 11, 1666 (2020) (see ref [49]). From [48] we can see that the results of our present manuscript are relevant for these reasons: 1) In [48] we employ the theoretical framework developed in our present work (sequential integration of Haar distributed unitaries) to analyze the trainability of the DQNN architecture. This shows that the tools we develop here can be employed to study other architectures. 2) Our results should be used to inform our intuition when designing cost-functions. In [49] the authors claim that their global cost function does not exhibit a barren plateau due to dissipative nature of the architecture. We show that this claim is incorrect as they do have an exponentially vanishing gradients. This goes to show that care should be taken when employing global costs, even in other architectures. 3) While the DQNN architecture is not an Alternating Layered Ansatz, in [48] we show that for specific DQNN architectures the results Corollary 2 can be applied and hence that trainability guarantees can be derived for DQNNs.
4. **Improved Presentation:** We have made the following changes to improve the presentation of our manuscript.
 - In order to improve the readability of the main text we have moved to the Methods sections previous Equations (7)-(18).

- We have added in the Methods a new figure (Fig. 6), and a new section entitled: “Intuition behind the main results”. There we provide some basic intuition behind our main results. This section is not meant to replace the “Sketch of the proof of the main theorems” but rather to present the reader with some insight regarding how the theorems are derived.
- To address the concerns of the third referee regarding Eq. (130) (current Eq. (132)), we have now added to the Supplementary Information an additional figure and associated discussion to show how this lower bound is obtained. Specifically we show that when O_i acts on m which are not the topmost qubits in $V_{\mathcal{L}}$, the prefactor one obtains is larger, and hence that the lower bound in Eq. (132) holds.

In our resubmission we have added the following files: Revised Manuscript, Revised Supplementary Information, Diff File for the Manuscript, Diff File for the Supplementary Information. In addition, as per requested we have also added a pdf containing the Main Text And Supplementary Information where we have highlighted the changes made.

In view of these improvements to our paper, we are resubmitting our manuscript “Cost-Function-Dependent Barren Plateaus in Shallow Quantum Neural Networks” to *Nature Communications*.

Sincerely,

Marco Cerezo, Akira Sone, Tyler Volkoff, Lukasz Cincio, and Patrick Coles

Our replies to the referee comments are below. The referee comments are quoted in italics.

I. Reply to Reviewer 1

Overall, my opinion on this article is that it is a "weak accept". For one, the authors study a regime that is significantly more non-trivial than that of [1], which also appeared in Nature Comm. This is because in the setting of shallow circuits, random circuits no longer form 2-designs and hence we cannot immediately rely on intuition from concentration of measure results in guiding our expectations. In addition to being technically and conceptually nontrivial, it is also certainly very timely as "NISQ devices" come online. In some sense, there is already a great deal of skepticism surrounding the "hardware efficient ansatz" which this paper studies, so from a practical standpoint these results may likely not be especially important in influencing the field.

Our main argument for considering the Hardware Efficient Ansatz (HEA) is that currently it is the best performing ansatz on NISQ hardware. This has been recently experimentally demonstrated by the Google group in current Ref. [7], where they show that employing a HEA leads to the best performance in an approximate optimization problem. For this application one could naively employ the Quantum Alternating Operator Ansatz (QAOA), but as shown in their work the large overhead of QAOA makes it significantly more sensitive to quantum noise.

In addition, even for quantum chemistry applications, state-of-the-art implementations do not employ ansatzes such as the Unitary Coupled Cluster (UCC) Ansatz. In fact, in Ref[8] also by the Google group, the authors employ an ansatz which has the same nearest-neighbor layered structure as the HEA we analyze in our manuscript. As the authors claim, “*The ansatz we consider affords us ways to minimize the resource requirements for VQE [...]*”.

In summary, these two state-of-the-art implementations for optimization and chemistry by the Google group actually employ hardware-efficient ansatzes similar to what we consider in our paper. We have now added a sentence in the introduction to remark this point. The sentence now read: “*Ideally, in order to reduce gate overhead that arises when implementing on quantum hardware one would like to employ a hardware-efficient ansatz [6] for $V(\theta)$. As recent large-scale implementations for chemistry [7] and optimization [8] applications have shown, this ansatz leads to smaller errors due to hardware noise.*”

Finally, let us remark that another advantage of the HEA is that it is problem agnostic, and hence very useful for applications where one does not possess any a-priori information (such as the case as the autoencoder algorithm considered in our heuristics as well as other quantum machine learning primitives).

– Despite the last bullet point in the "pros" column, I'm somewhat doubtful that these results will substantially influence the field. This is because this article largely gives negative results for the "hardware-efficient ansatz", which as far as I can tell most of the community has already become quite skeptical of. Personally, my main takeaway from this work is that this ansatz is even worse than many expected, in the sense that it already encounters the vanishing gradient problem at shallow depths for many reasonable choices of cost function. So, this paper gives nice theoretical results about how this ansatz is often bad, but many already considered it bad to begin with.

As mentioned in our previous reply, Hardware Efficient Ansatzes are still employed in state-of-the-art implementations (see Refs. [7-8]). In addition, let us remark that the theoretical tools we developed in our manuscript can be used to analyze the trainability of more general quantum neural network architectures such as the Dissipative Quantum Neural Network (DQNN) architecture proposed in current Ref. [49] (K. Beer et. al., Nature Communications 2020). In current Ref. [48] (K. Sharma, M. Cerezo, et al, arXiv:2005.12458) we have recently employed these theoretical tools to show that DQNNs can exhibit barren plateau which depend on the locality of the cost function. Hence, the results in this manuscript can inform our intuition for neural network architectures beyond the Hardware Efficient Ansatz.

With that being said, the main point of our work is not about ansatzes but rather pertains to cost functions. Namely, we investigate the properties of cost functions that lead to trainability. We identify locality of the terms in a cost function with trainability, and conversely we identify globality with the presences of barren plateaus. Hence, the analytical results of our manuscript should be considered as guidelines to designing cost functions that could be extended to other scenarios beyond the Hardware Efficient Ansatz.

In addition, in the context of chemistry or materials science, the cost function is typically determined by a transformation from fermionic operators to Pauli operators, and hence our work can inform researchers' choice about which transformation to use (i.e., Jordan-Wigner versus Bravyi-Kitaev). Namely, the transformation that leads to more local Pauli terms may also lead to a more trainable cost function, according to our results. We have added a remark in the Discussion section about the relevance of our work to choosing fermionic-to-Pauli operator transformations.

- I wasn't quite sure what to make of the "positive" result of Theorem 2/Corollary 2, which says that for shallow variational circuits and for a broad class of local, 1D cost functions, the gradient vanishes not exponentially with system size but rather as $\Omega(1/\text{poly}(n))$. This result feels rather weak to me, for a couple of reasons. (1) It seems we still encounter vanishing gradients in this case, in the sense that we'll still have $\text{norm}(\text{gradient}) - > 0$ with probability $- > 1$ as the system size is scaled up, albeit with polynomially-fast convergence rather than exponentially-fast.

We believe that Theorem 2 / Corollary 2 are exciting because they represent the first result that guarantees the trainability of quantum neural networks for *some* regime. In other words, we provide the first polynomially vanishing bound on the gradient for quantum neural networks.

Note that we use the term "*trainable*" when referring to a polynomially vanishing gradient. The referee notes that a polynomially vanishing gradient is still vanishing. However, we hope to shed some light on this issue in what follows, justifying why we use the term "trainable" for this sort of scaling.

Polynomially vanishing gradients are crucial for quantum speedup. That is because a gradient that vanishes as $1/\text{poly}(n)$ means that the algorithms precision should scale polynomially with n , i.e., the number of shots needed to estimate the gradient should grow as $\text{poly}(n)$. This polynomial overhead still allows for the possibility of exponential speed-up over classical algorithms (e.g., classical algorithms for training matrices of size $2^n \times 2^n$). Hence, in this regime, gradient-based optimization still has the potential for quantum speedup.

In contrast, exponentially vanishing gradients automatically destroy quantum speedup. In other words, if you have a barren plateau, then your algorithm will have exponential scaling. The standard goal of quantum algorithms is to avoid the typical exponential scaling that one has for classical algorithms. Hence, quantum algorithms (e.g., Shor's algorithm) typically aim for polynomial scaling. A variational quantum algorithm that has a barren plateau will need to employ an exponentially large number of shots to estimate the gradient. Consequently, this variational quantum algorithm will not yield the exponential speedup ("quantum speedup") that one typically desires for quantum algorithms.

In summary, the difference between polynomially vanishing and exponentially vanishing gradients is at the heart of quantum speedup. If your variational quantum algorithm has a polynomially vanishing gradient, then it still has hope of providing a quantum speedup over a classical algorithm.

We have now added a paragraph in the introduction that reads: "*We remark that polynomially vanishing gradients are crucial for quantum speedup. A gradient scaling as $O(1/\text{poly}(n))$ means that the number of shots needed to estimate the gradient should grow as $O(\text{poly}(n))$. In contrast, exponentially vanishing gradients automatically destroy quantum speedup, i.e., the presence of barren plateaus implies that the algorithm will have exponential scaling. Noting that the standard goal of quantum algorithms is polynomial scaling as opposed to the typical exponential scaling of classical algorithms, a VQA with a polynomially (exponentially) vanishing gradient has hope (no hope) of achieving this goal. Hence we employ the term "trainable" for polynomial scaling.*"

(2) Moreover, this holds only in the rather weak setting of $O(\log n)$ -depth circuits with 1D, spatially-local cost functions. So, while I certainly found this result mathematically interesting, I actually viewed it more as a negative result for variational algorithms, rather than a positive result as the authors billed it as. Maybe I'm not fully appreciating something here - perhaps the authors could provide a bit more motivation for this result.

We now respond to the referee’s comment that we treat a “rather weak setting of $O(\log n)$ -depth circuits with 1D, spatially-local cost functions”.

In our revised manuscript, we have now generalized our results from 1D connectivity and shown that our theorems hold in 2D grid-like connective such as the one employed in IBM’s and Google’s quantum devices. This is a non-trivial generalization that makes our revised manuscript significantly stronger than the previous version. Consequently, we have modified Figure 3, and in the Ansatz section we have added the following sentence: “*Note that, as depicted in Fig. 3(c), the Alternating Layered Ansatz can be readily implemented in devices with one-dimensional connectivity, as well as in devices with two-dimensional connectivity (such as the one found in IBM’s [34] and Google’s [35] quantum devices).*”

We also remark that spatially-local cost functions encompass a wide range of interesting applications for variational quantum algorithms. For instance, for many VQE applications the operator O corresponds to a Hamiltonian which is usually expressed as a summation of local Pauli terms. Such is the case for certain k -local Hamiltonians. In the more general setting, when the operator O contains local and global terms our results show that the local terms will be trainable. Hence, as previously mentioned, this can be useful when designing algorithms. For example, in quantum chemistry, when transforming a fermionic Hamiltonian to a spin Hamiltonian one can employ either the Jordan-Wigner or the Bravyi-Kitaev transformation. Since each of those leads to a different number of local terms in the bosonized Hamiltonian, one can choose which transformation to use to get better trainability guarantees. This result has been in fact verified in current Ref. [46] where the authors employ our results to note that: “*For Bravyi-Kitaev mapping, the barren plateau regime set on more gradually.*” We have added now the paragraph sentence in the discussion section that reads: “*In the context of chemistry or materials science, our present work can inform researchers about which transformation to use when mapping a fermionic Hamiltonian to a spin Hamiltonian [44], i.e., Jordan-Wigner versus Bravyi-Kitaev [45]. Namely, the Bravyi-Kitaev transformation often leads to more local Pauli terms, and hence (from Corollary 2) to a more trainable cost function. This fact was recently numerically confirmed [46].*”

Finally, we also note that trainability guarantees of $\log(n)$ -depth Hardware Efficient Ansatzes provides several different interesting settings that can be further explored. For instance, we have also added a paragraph to the discussion section which reads: “*Moreover, the fact that Corollary 2 is valid for arbitrary input quantum states may be useful when constructing variational ansatzes. For example, one could propose a growing ansatz method where one appends $\log(n)$ layers of the hardware-efficient ansatz to a previously trained (hence fixed) circuit. This could then lead to a layer-by-layer training strategy where the previously trained circuit can correspond to multiple layers of the same hardware-efficient ansatz.*”

– *On a similar topic as the above complaint, the numerical experiment did show a speedup for the "local cost function" case over the "global cost function" case, but the circuit depth for this experiment was only four. It seems plausible that even the "local" case might quickly become untrainable as the depth is increased. It would have been interesting for the authors to also try to understand how trainability scales with depth in the local case for this model.*

We thank to the referee for their comments. We have now added Figure 6 which shows how trainability scales with the number of layers in the Alternating Layered Ansatz. IN this figure we show the dependence on L when training with our proposed local cost for the autoencoder example with $n_A = 1$ and $n_B = 10$. As one can see, the training becomes more difficult as L increases. Specifically, as shown in the inset it appears to become exponentially more difficult, as the number of shots needed to achieve a fixed cost value grows exponentially with L . This is consistent with (and hence verifies) our bound on the variance in Theorem 2, which vanishes exponentially in L . We have also added a paragraph analyzing these results in the Numerical Simulations section.

– *While the paper is well-written, one downside is that the authors do not provide much intuition for many of their results, instead only giving long brute-force calculations. I appreciated the toy model demonstrating the vanishing gradient phenomenon for shallow circuits with a non-local cost function, but many other results did not have such accompanying intuitive explanations.*

We have now added a section in the Methods section called “*Intuition behind the main results*”. Therein we show how the variance of the cost function vanishes exponentially with mL for local cost functions, while for global costs the variance vanishes exponentially with n even in the one layer case. While this section is not intended to replace the main proofs we believe it will provide the reader some intuition behind our theorems. This section now also bridges the gap between the warm-up example and our main theorems and also showcases the tools used to derive our main result.

More detailed comments/corrections: – In Eq. (8), I think you mean for V_L and V_R to be swapped. This same mistake occurs throughout the document.

- Typo in p. 4: "bistrings" -> "bitstrings"
 - In p. 9 and p. 13, you define a unitary t -design to be a finite set, however continuous distributions can also be t -designs. Maybe you could instead word this as something like, 'this finite set is a unitary t -design if...'
 - In p. 14 Eq. 53, I think the 'F' should be a 'D'. Also, $V_y \rightarrow W_y$.
 - Typo in the expression for A_{qp} in Eq. (58).
 - In p. 18 Lemma 7, O_1 should be O_2 . Also $\text{Tr}_k[O] \rightarrow \text{Tr}_j[O]$.
 - Unless I'm misunderstanding something, I think there may be a (noncritical) bug in your proof of Proposition 1 in Supplementary Note 2. In particular, you say "the radius of the integration ball vanishes" when $n \rightarrow \infty$. But it appears to me that the radius approaches $-\ln(1 - \delta) > 0$ in the limit. If I'm correct about this, this proof should be revised.
 - In p. 21 Eq 98, I think you actually mean $V_L \rho V_L^\dagger$
 - In p. 22, when you reference Eq 103 I think you might mean Eq 102.
- [1] McClean, Jarrod R., et al. "Barren plateaus in quantum neural network training landscapes." *Nature Communications* 9.1 (2018): 1-6.

We thank the referee for their corrections. We have made appropriate changes in the manuscript to address these issues.

II. Reply to Reviewer 2

For clarity and to avoid ambiguity, we suggest to lean away from the term QNN. It is never used beyond the title and the abstract. We suggest replacing QNN with parameterized quantum circuits (PQCs), which in our opinion could be broader since the results extend beyond ML applications. If the authors want to stick to the term QNN they should motivate it more and even define what are the requirements of a circuit to be a QNN, something that apparently there is no consensus in the literature. The QNN term does not add any additional "IJML coolness", and on the contrary, it can cause ambiguity, since not all PQCs and/or not all problems with hardware-efficient ansatz are QNNs.

We thank the referee for their suggestion. We here present four arguments for why we believe that using the term Quantum Neural Network (QNN) is appropriate. First, note that it is now standard for the community to use the term QNN when referring to Parametrized Quantum Circuits (PQCs). For instance, the *Nature Communications* article in [1] employs the term QNN when working with PQCs. Hence, we are simply employing commonly used terminology. Second, using QNN can also be justified from the fact that neuron activation functions can be embedded inside unitary gates (see [2]). Third, note that the main numerical implementation performed in our manuscript was for quantum autoencoders, and it is standard practice to view such autoencoders as neural networks. Because this is our main illustrative example, we wanted to draw the attention of the quantum machine learning community to our manuscript with the term QNN. Finally, as shown in our recent work [3], the results obtained in our current manuscript can be directly applied to other architectures that have already been called Quantum Neural Networks in the literature (see [4]). Hence, we believe that it is appropriate to employ the term QNN in our manuscript.

The authors provide a derivation of their asymptotic bounds for which they introduce a graphical calculation method using tensor networks and we agree the derivation of the theorems and corollaries are rigorous. And even though the numerical example clearly illustrates the phenomenon, the main claim about "trainability" of the parametrized quantum circuits (PQCs) is not well justified. Success in the trainability of PQCs is not equivalent to existence or prominence of barren plateaus.

Let us first remark we have justified the use of the term "trainability" in a newly added paragraph in the Introduction of our manuscript. In addition, this justification appears above in our reply to the first referee, which we recall as follows.

We use the term "trainable" when referring to a polynomially vanishing gradient (and hence to the case when there are no barren plateaus). In what follows we hope to justify why we use the term "trainable" for this sort of scaling.

Polynomially vanishing gradients are crucial for quantum speedup. That is because a gradient that vanishes as $1/\text{poly}(n)$ means that the algorithms precision should scale polynomially with n , i.e., the number of shots needed to estimate the gradient should grow as $\text{poly}(n)$. This polynomial overhead still allows for the possibility of exponential speed-up over classical algorithms (e.g., classical algorithms for training matrices of size $2^n \times 2^n$). Hence, in this regime, gradient-based optimization still has the potential for quantum speedup.

In contrast, exponentially vanishing gradients automatically destroy quantum speedup. In other words, if you have a barren plateau, then your algorithm will have exponential scaling. The standard goal of quantum algorithms is to avoid the typical exponential scaling that one has for classical algorithms. Hence, quantum algorithms (e.g., Shor’s algorithm) typically aim for polynomial scaling. A variational quantum algorithm that has a barren plateau will need to employ an exponentially large number of shots to estimate the gradient. Consequently, this variational quantum algorithm will not yield the exponential speedup (“quantum speedup”) that one typically desires for quantum algorithms.

In summary, the difference between polynomially vanishing and exponentially vanishing gradients is at the heart of quantum speedup. If your variational quantum algorithm has a polynomially vanishing gradient, then it still has hope of providing a quantum speedup over a classical algorithm.

The proofs and the results are binded to the strategy of randomized initialization AND the usage of the gradient as a training method since the assessment is made by calculating their variance. Therefore, no conclusions can be made about the trainability of the PQC, for example, under a black-box solver, or any other type of solver or computational strategy which exploits the structure of the problem.

Let us now address the referee’s claim that our results are only applicable to the trainability with gradient-based training methods. As we argue below, gradient-free methods (black-box solvers) will also not work in the presence of barren plateaus. Furthermore, we emphasize that our numerical implementations were actually performed with a gradient-free optimization method (see Optimization section in Methods). As one can see from our numerical results, the global cost function for the quantum autoencoder example is completely untrainable for large system size with the gradient-free optimizer that we employ.

As we now argue, the existence of a barren plateau implies that any optimization method (even black-box optimizers) will fail unless the algorithm has exponential scaling. For this, let us first remark that the barren plateau phenomenon provides information about the cost function landscape and is not associated with any particular method one can employ to train said cost. In particular, when the variance of the cost function partial derivative vanishes exponentially with n , then in average the cost landscape will be essentially flat (as the slope is of the order $O(1/2^n)$).

Let us now discuss how barren plateaus can affect gradient-based, and gradient free-optimization methods. As previously discussed in this reply, when employing a gradient-based optimizer, then the algorithm’s precision in estimating the gradient should grow as $O(2^n)$. Hence, this means that the algorithm is inefficient as it has exponential scaling.

For instance, let us closely analyze how barren plateaus make gradient-free black-box optimizers, such as the Nelder-Mead simplex method. For the clarity of discussion let us briefly review this algorithm. Given a D -dimensional space one can build a simplex consisting of $D+1$ points: $x_1, x_2, \dots, x_{(D+1)}$. Then, let $C(x)$ be the function we are attempting to optimize. At each iteration the Nelder-Mead takes the following steps:

I. Ordering. All the points are ordered/sorted, such that the value of C for the first point is the lowest, and that for the last point is the highest. Let the indices of the first(worst), second(second worst) and last(best) points be h, s, l respectively.

II. Computing the Centroid. Consider all points except the worst (x_h), compute the their (mean).

III. Transformation. In this step the simplex is transformed in order to find a minimizing direction. The simplex can then be: reflected, expanded, or contracted.

Note that in a barren plateau regime, the cost function value for any of the initial points will be essentially the same (up to exponentially small deviations), i.e., $C(x_1) \approx C(x_2) \approx \dots C(x_{D+1})$. Hence, in order to successfully order these values the algorithm needs exponential precision in determining the cost function values. If one does not have exponential number of shots, then the precision in estimating each $C(x_i)$ will be given by the shot noise, and the points will be randomly ordered. Hence, the transformation in step III will essentially be a random walk. The latter then shows that the Nelder-Mead will fail unless the algorithm has exponential scaling.

We have now added this sentence in the Optimization section in the methods section: “*Finally, let us remark that while we employ a sub-space search algorithm, in the presence of barren plateaus all optimization methods will (in average) fail unless the algorithm has a precision (i.e., number of shots) that grows exponentially with n . The latter is due to the fact that an exponentially vanishing gradient implies that in average the cost function landscape will essentially be flat, with the slope of the order of $O(1/2n)$. Hence, unless one has a precision that can detect such small changes in the cost value, one will not be able to determine a cost minimization direction with gradient-based, or even with blackbox optimizers such as the Nelder-Mead method [54].*”

As pointed in the paper, there have been many strategies proposed to make circuits with barren plateaus trainable and to cope with this issue. Some of these being initialization strategies, meta-learning heuristics, layer-by-layer training,

and some others that have come in part as motivation from the original paper by McClean et al.. It would be good to see if the claims from the authors hold even with black-box solvers. At least this is something outside the scope of the theorems derived and therefore the manuscript should be revisited throughout to avoid any claims related to trainability and their connection to the locality of the cost function. From their main results and theorems, the authors can claim that there is a link with the locality of the cost function to the presence or absence of barren plateaus, but saying that this separates a PQC design as trainable or not trainable could be simply wrong or at least certainly beyond the scope of this work.

As noted above, our work also has relevance to black-box solvers such as Nelder-Mead. Hence we do not believe that such solvers will be able to avoid the exponential overhead associated with barren plateaus.

The referee also mentions other strategies that have very recently been proposed in the literature in order to address the barren plateau issue. While some of these strategies seem promising, we do wish to emphasize the distinction between a rigorously proven result (such as the results in our manuscript) and a heuristic strategy.

The main goal of these heuristic strategies is to essentially change the cost function landscape (or initialize to a different part of the landscape) so that it no longer presents a barren plateau. Each strategy needs to be individually analyzed, as they will in general fall outside the scope of our main results (which hold for randomly initialized layered hardware-efficient ansatzes). Hence, Theorem 2 (and specifically Corollary 2) of our manuscript are the only known rigorous results that can provide trainability guarantees, and a clear strategy for avoiding the exponential scaling induced by barren plateaus. Hence, we believe that our work paves the way towards more rigorous scaling results which are also desperately needed for these heuristic strategies. Our work can therefore be viewed as motivating a new direction of research of rigorously proven trainability guarantees.

As the authors claim that trainability is a function of the locality of the cost function observable, they need to provide a discussion on how to reconcile the conclusion of this paper: <https://www.nature.com/articles/s41467-020-14454-2>, with their conclusions. In this reference, the authors define a QNN with a global cost function and they claim “absence of a barren plateau in the cost function landscape.” They go further and mention just before the discussion section that the cost function is non-local means and in the words of the authors, “this means that Levy’s lemma-type argumentation does not directly apply.”

Let us first remark that the Barren Plateau result of current Ref [9] (McClean et al. Nature Communications 2018) is not directly derived from the concentration-of-measure and Levy’s Lemma. Rather, their result is a statement about the variance of the cost function partial derivative which holds when the parameterized quantum circuit form a 2-design on n -qubits. Such result is valid for both global and local cost functions, as in their manuscript the authors make no assumption regarding the locality of their Hamiltonian H . The fact that the result in [9] is not about Levy’s Lemma can be seen from the proof of their main results: the authors do not consider a reduced subsystem of the state obtained from the parameterized quantum circuit (which is the typical concentration-of-measure setting).

We give further mathematical details on why the results of [9] hold for both global and local cost functions, as follows. From Eq (7) of [9] we know that if U_+ and U_- form independent 2 designs, then $\text{Var}[\partial C] \sim \text{Tr}[H^2]\text{Tr}[\rho^2]\text{Tr}[V^2]/2^{3n-1}$. Assuming that V is a Pauli operator on a single qubit, then $\text{Tr}[V^2] = 2^n$. Then, let us recall that for any quantum state we have $\text{Tr}[\rho^2] \leq 1$. Under these assumption, Eq (7) in Ref [9] leads to $\text{Var}[\partial C] \leq \text{Tr}[H^2]/2^{2n-1}$. Let us assume that $\text{Tr}[H^2] \leq 2^n$, which holds for the global and local cost functions considered in our manuscript. Then we have $\text{Var}[\partial C] \leq 1/2^{n-1}$. which implies that under these assumptions the barren plateau in [9] holds for local and global Hamiltonians.

We thank the referee for pointing us towards current Ref. [49] (K. Beer et. al., Nature Communications 2020), which presents a very interesting architecture for QNNs. As the referee correctly points out, the authors of [49] indeed claim that their dissipative architecture (which employs a global cost) should not exhibit barren plateaus in the cost function landscape. Since it is not straightforward to show that the result in [9] applies for this dissipative architecture, we have fully analyzed the trainability of the dissipative quantum neural network of [49] in a manuscript which we have recently uploaded to the ArXiv (see Ref [48], (K. Sharma, M. Cerezo, et al, arXiv:2005.12458)). This work is an extension of our present manuscript where we employ the same theoretical framework (sequential integrating of Haar-distributed unitaries).

We remark that this shows the power of the techniques developed in our current manuscript, as these techniques apply to other ansatzes and architectures. We have now added a sentence in the Discussions where we remark this fact. The sentence reads: “*Finally, we emphasize that while our theorems are stated for a hardware-efficient ansatz, it remains an interesting open question as to whether other ansatzes and architectures exhibit similar scaling behavior as that stated in our theorems. For instance, it has been recently shown [48] that our results can be extended to a more general type of QNN called dissipative quantum neural networks [49].*”

In [48] we show that the claim regarding the “absence of a barren plateau in the cost function landscape” is incorrect as the QNN of [49] does exhibit barren plateaus when employing a global cost function. The results that we find in [48] are analogous to those in Theorem 1 / Corollary 1, as we find that global costs exhibit barren plateaus even for shallow circuits. In [48] we also find that Theorem 2 / Corollary 2 of our current manuscript can be immediately applied to the architecture in [49], and hence we provide some trainability guarantee for the architecture in [49], provided a local cost function is employed.

III. Reply to Reviewer 3

In this paper, the authors discuss trainability of quantum neural networks at the initial stage of the training. This kind of problem has been investigated in the field of classical neural network as well, for example in [Pennington et al.]. This or other related papers/conference proceedings can be quoted. With classical or quantum neural networks, we first construct networks randomly and improve the parameters of the networks to reduce the value of a chosen cost function (or loss function). This is where, random matrix theory plays a role. To make the value of the cost function smaller, we use gradients of the cost function with respect to the parameters that we improve. The authors discuss this matter in terms of choice of cost functions. More precisely, their conclusions state that local cost functions are better than global cost functions with Variational Quantum Algorithms (VQAs) because gradients vanish slower with local cost functions for randomly constructed Alternating Layered Ansatz. To this end the authors utilized Weingarten Calculus, but they should quote [Collins et al.], where they provided exact Weingarten formulas for unitary, orthogonal and symplectic groups. The results of this paper should be known widely because QVAs have a lot of applications and their models have counterparts in the real world. However, the mathematical discussions have not convinced me yet. (I might be looking over something.) I shall start with the question that I think are important.

We thank the referee for point out these references. We have appropriately cited them in our paper.

In the proof of theorem 2 of Supplementary Note 6, the authors use special case where O_i and O_j act only on topmost m qubits in $V_{\mathcal{L}}$. In this case, as the authors wrote, only $L-l$ blocks are non-trivial, and special iterative calculations are possible as in Sup. Fig. 1. However, in general we get after the integration over W four different terms for each, like in Lemma 6. This is why, I have two questions about (130) when such a special case does not hold.

1) Integrating over more of W s, how does it affect the coefficient? How the power $L-l$ changes?

Let us first remark that in the previous version of our manuscript we had only included the case when the operator O_i acts non trivially on the topmost m qubits in $V_{\mathcal{L}}$, as for other cases we could not obtain a closed expression for $T_{\tau} := \sum_{\tau} \hat{t}_{\tau}^{ii}$. However, we have now added a new figure (and associated discussion) in the Supplementary Information to address the case when O_i acts non trivially on a different set of qubits. Essentially, we show that this case leads to a larger T_{τ} .

As shown in our new Sup Fig 2(a) and (b), we compare two cases:

- Case 1: when O_i acts non trivially on the topmost m .
- Case 2: when O_i acts non trivially on S_w (i.e., on the “middle qubits”).

In the second case the number of gates one has to integrate is $(L-l)(L-l+2)/4$. While Case 1 was analyzed in Sup Fig 1, and in current Equation (132), we explicitly calculate the value of T_{τ} for $(L-l) = 2, \dots, 5$. These results were obtained by explicitly integrating over each block and are now presented in Eq. (138).

In Sup Fig 2(c) we then compare the values of T_{τ} for Case 1 and Case 2 versus m , and we show that:

- For all cases analyzed, the value of T_{τ} from Case 2 is larger than the value for Case 1.
- For a given $(L-l)$ the difference between the T_{τ} in Case 1 and Case 2 scales as $1/2^{m(L-l-1)/2}$, meaning that as $(L-l)$ (or m) increases the inequality (135) becomes tighter.
- As shown in panel (c) of Sup. Fig. 2, the coefficients T_{τ} in Case 1 for $(L-l)$ converge to the values of Case 2 for $(L-l-1)$, which means that one can always estimate the values of T_{τ} for Case 2 from the values in Case 1.

These results can be understood from two key facts: First, that while integrating over a block leads to coefficients smaller than one according to (56), one also gets more contributions for $\epsilon(\hat{O}_i)$. Hence, integrating over more blocks lead to more terms proportional to $\epsilon(\hat{O}_i)$. Second, we note in Case 2 the integration over the last blocks will lead to multiplicative factors $2^{m/2}$ (which essentially come from taking the trace over the identity over $m/2$ qubits). In Sup. Fig. 2(a) we have schematically indicated by arrows the indexes that will lead to delta functions in \mathbf{p} and \mathbf{q} after each block is integrated. On the other hand, in Sup. Fig. 2(b) we have also indicated the indexes that will lead to factors $2^{m/2}$.

II) After such integration over a W , we usually we get terms with negative signs as in Lemma 6. How did the authors handle these negative terms appearing for each integration over a W ?

As correctly pointed out by the referee, each integration of a block W leads to four terms, two of which have negative coefficients. However, as shown in Lemma 6, integrating a term which has the form of $\Delta\Omega = \text{Tr}[AA'] - \text{Tr}[A]\text{Tr}[A']/d^2$ leads to new terms that have exactly the same form as the original $\Delta\Omega$ (i.e. to terms of the form $\text{Tr}[BB'] - \text{Tr}[B]\text{Tr}[B']/d^2$). Moreover, we remark that the terms $\Delta\Omega$ take into account the negative contribution. Hence, one can recursively employ Lemma 6, and one only needs to keep track of how the traces and the operators B and B' in the new $\Delta\Omega$ are defined. We refer the referee to Sup. Fig. 1, where we schematically show this procedure.

These questions are important because a lower-bound is derived in (130).

Besides, I am wondering how (140) average of Hilbert-Schmidt norm was calculated. Explanations should be added because usually square root does not get along with integration.

We thank the referee for this comment as we noted that we were lacking in our manuscript the definition of the Hilbert-Schmidt norm. In our manuscript we define D_{HS} as $D_{HS}(\rho, \sigma) = \text{Tr}[(\rho - \sigma)^2]$, and hence no square root arises. In order to avoid confusions we have recalled this definition throughout the manuscript.

*There are several comments on Lemma 6. First, notations are confusing. Apart from the typo: $d = d * d * d * d = d^4$, the authors should state what are x and y , and which are $\hat{A} \hat{I} \hat{J} \hat{I} 2$ possible cases $\hat{A} \hat{I}$. The notations like $H_4 \in H_k$ after (65) is confusing. The proof is confusing to me too. First, the equation (71) seems strange because the LHS is the average over W s but the RHS still has W s in Ω s. Perhaps, the authors wanted to write another equation. Second, the authors tried to $\hat{A} \hat{I} \hat{J} d$ deduce (71) $\hat{A} \hat{I}$ in the proof by using the statement in the lemma to be proved. This structure should be avoided.*

We thank the referee for pointing out these issues. We have modified the wording in Lemma 6 in order to improve readability and clarity. Specifically, we now employ the more conventional notation $H_4 \leq H_k$ to indicate that H_4 is a subspace of H_k . We have also fixed previous Eq. (71), where we replaced W by A . Finally, we have removed previous Eq. (71) from the proof as this is just a particular example of the Lemma we later employ in the Supplemental Information.

In the proof of Proposition 1, "the radius of the integral ball vanishes", but it seems to me that the limit of $n((1 - \delta)(1/n))$ is $-\log(1 - \delta)$ and it does not vanish.

We thank the referee for this remark, We have now fixed this non-critical bug in the proof.

Moving to Supplementary Note 4, and my question is what if $\nu = 1$. The result holds if W_A and W_B are both 2-design, but $\nu = 1$ implies that W_B is the identity from (87).

We note that the approximation that W_A and W_B are independent 2-designs is a necessary one for deriving our main results. In fact, since both W_A and W_B form 2-designs we find that the variance general formula is "symmetric" for $\Delta\Psi$ and $\Delta\Omega$, and this leads to an easier derivation of our theorems.

While this is indeed an approximation, we remark however that the our numerical implementation verify our main results even if we employ an ansatz where W_A , W_B , nor any block W forms a 2-design. The latter entails that albeit the approximations we make our theorems, our main results still inform our intuition even in scenarios where they don't directly apply.

I shall make some minor comments in the rest of letter. Is the order correct in (8)? In addition, V_R should be defined. In (58), there is a typo. There are typos in Lemma 7 (the second O_1 should be O_2 ?) and in Proposition 1 (θ is missing in the definition of A_S^G). trace is missing in (103). I found several S "topomost" instead of "topmost".

We thank the referee for pointing out these typos. We have addressed and fixed them.

Reviewers' Comments:

Reviewer #1:

Remarks to the Author:

I thank the authors for addressing the points raised. Due to time constraints I wasn't able to carefully proofread all the technical details in the Supplemental Material, but I do have some additional followup concerns/questions. Besides the following concerns, I am happy with the current version of the manuscript.

1. The authors have fixed a bug in the previous version of the manuscript which has changed Proposition 1, which analyses the "narrow gorge" phenomenon for a toy problem. However, it seems to me that the current version of Prop. 1 is quite weak. The current version shows that with high probability, the global cost function is delta-far from the optimum in cost function value, for $\delta = 1 - (1 - c/n)^{1/n}$ with $c = O(1)$. However, asymptotically this scales like $\sim c/n^2$ which vanishes as n increases. So for example, Prop. 1 doesn't even rule out the possibility that the global cost function asymptotically approaches the optimum (like $\sim 1/n$ for example) with high probability when randomly initialized. Based on the explanation of the narrow gorge phenomenon, I would expect that with overwhelming probability, the global cost function value will be close to 1, which is much stronger than what Prop. 1 shows. Conversely, for the local cost function C_L , it is shown that the cost function is delta-close to the optimum with high probability if $\delta > 1/2$. Hence, the Prop. fails to prove that C_G and C_L behave much differently for this toy problem. For example, as far as I can tell the following possibility is not ruled out by this result: the distributions of C_L and C_G may be asymptotically identical as $n \rightarrow \infty$, concentrating around the value $C_L = C_G = 1/4$. Since the purpose of the Prop. is to establish significantly different behavior of C_L and C_G , I feel the result should be strengthened to rule out the sort of example above.

2. I'm confused about the extension of the results to 2D. Referring to Fig. 3c, from the diagram it looks like the extension of the ansatz to 2D simply consists of grouping sites together to effectively collapse the 2D grid into a 1D chain, which can subsequently be analyzed as in the original 1D case. Indeed, you say "we remark that the following results are valid for both cases as the lightcone structure will be the same". However, I am struggling to understand how this picture is motivated. In a 2D $\sqrt{n} \times \sqrt{n}$ square grid of qubits, each block would then need to consist of at least \sqrt{n} qubits to collapse to the 1D case, which does not seem well motivated, since each block needs to form a 2-design. Also, the lower bound of Theorem 2 becomes uninteresting if $m \sim \sqrt{n}$. It seems like the more natural setting in the 2D case would involve each block involving only a constant number of sites, and the blocks would be tiled in such a way to form a natural circuit architecture. However, in this setting the lightcone structure is not obviously equivalent to the 1D case, so I don't see how the 1D arguments are valid. Perhaps I'm just confused about your argument, in which case some extra discussion about how the 2D case reduces to the 1D case may be helpful. If you have in mind that one of the dimensions of the 2D grid be held constant so that $m = O(1)$, then I feel this should not be referred to as a 2D result as it effectively is still a 1D ansatz (it is just a way of implementing the 1D Alternating Layered Ansatz). Indeed, in this case I don't understand what the purpose is of drawing a distinction between 1D and 2D implementations of the (ultimately 1D) Alternating Layered Ansatz.

3. I think Theorem 2 (the lower bound) is either technically incorrect as stated or there is some notational ambiguity, as there appears to be a simple counterexample. Consider the circuit consisting of just one block of the form $\exp(-i\theta P)U$ which forms a 2-design, where P is a Pauli and U consists of some other unitaries. Suppose the target operator O is simply $O = P$. Then the cost function is $C(\theta) = \langle 0|U^\dagger \exp(i\theta P) P \exp(i\theta P) U|0\rangle = \langle 0|U^\dagger P U|0\rangle$. Clearly this is independent of θ , so the gradient of C is zero everywhere (and hence the variance is zero everywhere). However, Theorem 2 as written predicts a positive variance. Presumably this is just some sort of edge case which can be easily accounted for, but Theorem 2 should be somehow modified to rule out this counterexample (and other possible edge cases). This will happen more generally when we compute the derivative with respect to a rotation in the last

layer which commutes with the operator O . Maybe Thm. 2 is not applicable for the last layer of blocks?

....

Minor:

- page 1. "Ideally, in order to reduces"  "reduce"
- page 3. "L layers of m-qubit gates". Should "gates" actually be something like "unitaries" since each block will generally consist of multiple individual gates?
- Fig. 3b. Missing a prime on O_i .
- page 5. "becomes trivial when the number of layers L is $O(\text{poly}(n))$ ". Should this technically be something like $\Omega(\text{poly}(n))$ or just $\text{poly}(n)$, since $O(\text{poly}(n))$ would generally be understood to include $\text{polylog}(n)$ for which your expressions are non-trivial?
- page 6. "dimensionallity" -> dimensionality
- page 11. "later"  latter
- page 13. Missing period at end of Section VII.

Reviewer #2:

Remarks to the Author:

Report to the authors:

We feel the authors need to tone down and add clarity in the manuscript to all the major points raised in this report before it is considered further for publication. In the current form, the manuscript is still very misleading to the scientific community interested in NISQ algorithms.

Major revisions suggested:

1) Use of QNNs in the body of text and title:

In their reply the authors provide four arguments of why using the terminology of QNNs is appropriate and interchangeable with the notion of parametrized quantum circuits. We strongly disagree with most of them and it looks the authors feed the confusion of what a machine learning task is about and ignore the minimum requirement for what constitutes a neural network, either classical or quantum, independently of the many quantum proposals out there. To address the comments from the authors, we paste here their reply for convenience:

First, note that it is now standard for the community to use the term QNN when referring to Parametrized Quantum Circuits (PQCs). For instance, the Nature Communications article in [1] employs the term QNN when working with PQCs. Hence, we are simply employing commonly used terminology. Second, using QNN can also be justified from the fact that neuron activation functions can be embedded inside unitary gates (see [2]). Third, note that the main numerical implementation performed in our manuscript was for quantum autoencoders, and it is standard practice to view such autoencoders as neural networks. Because this is our main illustrative example, we wanted to draw the attention of the quantum machine learning community to our manuscript with the term QNN. Finally, as shown in our recent work [3], the results obtained in our current manuscript can be directly applied to other architectures that have already been called Quantum Neural Networks in the literature (see [4]). Hence, we believe that it is appropriate to employ the term QNN in our manuscript

These arguments are flawed in many ways and indicate a lack of understanding from the authors about even what is machine learning and what quantum neural networks are used for. Although some QNNs happen to be PQCs (the reason why the authors tend to believe the community is using the term QNN for PQCs), not all PQCs are QNNs. It is important to note that all or most of

the papers that have used the term of QNN to denote PQC are used in the context of machine learning tasks. For a learning task to be valid, one needs to have the concept of a training data set from which a learning task can be defined in the form of a cost function that measures the performance of a trained neural network. Examples of these QNNs based in PQC are abundant and range from classification/supervised learning tasks and generative unsupervised learning tasks. Referring that the community is now using QNNs to refer to PQC is misleading as it reinforces a confusion between distinct concepts. This is the basis of us requesting the authors change the title to be more precise to what their results are about. Since the authors themselves seem to want to sell their results as QNNs, it is fine to mention the connection of PQC to QNNs inside the text, but they need to clarify this distinction.

It is interesting that in their argument they refer to the paper of McClean et al. published in Nat. Comms (Ref. [1]) and where the first results about barren plateaus were presented. Although it was used by the authors in the title, in reality QNNs are not mentioned anywhere in the text. We counted 1 occurrence on the work QNN in that paper, while about 100 times they used the expression random parametrized quantum circuits (RPQC). We feel that RPQC is a good description of what the theory inside this paper from Cerezo et al. is about, and that even using that term is generous since the theorems and results of this paper under review deals with a narrower scope: that of the layered version of the RPQC which requires a 2-design per block units in the layers. We also note a recent publication from the same group of authors here, <https://arxiv.org/abs/2007.14384>, use the term VQA in the title and avoid QNN. We feel that is already a more sensible term to be used in the title than QNN.

To summarize. Although some PQC have been used to construct some QNNs proposals, it is important these are taken into the right context of learning tasks and not used interchangeably as the authors argue and use it in the text (for example, ground state preparation in chemistry or QAOA are not learning tasks since there is not a training set. But these both are VQA and PQC). As long as the PQC is used for a learning task, then QNN is a good term to be used there. Therefore, although the results here would be applied to that family of QNNs or some related ones which are based in layered RPQCs, saying that QNN are PQC are being used interchangeably in the community is wrong and unjustified. Any paper using QNN, if a decent one, at a minimum is solving either a supervised, unsupervised or reinforcement learning task. If this abuse of language has been used by some authors, this practice needs to stop for scientific integrity and correctness.

2) Usage of terms such as trainability and trainability guarantees:

The authors argue that they added a paragraph to justify the term trainability in the introduction. First, it is important to note the paragraph as written is misleading (see section below on quantum speedup for more details). Aside from the need for that to be addressed and corrected, the paragraph is the last one in the introduction while the authors refer to trainability several times before in the abstract and introduction. The definition of what is meant by trainability and trainability guarantees should be stated up front the first time the term is used. This is critical to the readability of the paper given that the definition used by the authors is unique and only applies to this paper. As we stated in our first report, saying a PQC with glocal cost function is not trainable is misleading since there are heuristic strategies which might be successful and therefore disproving the trainability claims. Furthermore, instead of deviating the attention to claims on quantum speedup, which are secondary, the most important statement in the introduction and in the definition is to remark the definition itself. Basically, a clear statement that trainability as used in the paper is latched to the main assumptions in the proofs: 1) random initialization of parameters, 2) the specific details of the layered ansatz proposed which is not hardware-efficient (more on this below), and 3) the explicit form of the cost function which does not necessarily capture all cost functions used in the literature (e.g., in several QML papers the KL divergence is widely used and it not of the form proposed since it has a logarithm of the probability. Since the proofs don't mention these cases, it should be stated that the results apply to a family of local and global cost functions, but the most generic case of arbitrary cost function is not covered (or if the

proofs cover any possible loss or cost function it would be to state that as well). We feel for the scientific integrity of the results presented, the authors should make an effort to state that saying anything beyond the assumptions in the proofs concerning this paper is beyond the scope and left as open questions. And this should be clear from early in the introduction before big claims about trainability are presented, and without clearly defining what is meant by that. It is a good point to remark that there are proposals to bypass these limitations, as the case of the proposals to deal with parameter initialization. We feel this is a more honest and needed clarification since as written now, it looks like the authors' claims on the trainability of PQCs, are more general than what they really are.

3) New claims about quantum speedup:

As pointed out by the authors, they added a paragraph at the end of the introduction on quantum speedup. Again, the authors communicate this very poorly and in misleading way by stating "vanishing gradients are crucial for quantum speedup". This is far from true for many reasons, First of all, there are more proofs on quantum speedup that are not even related to training PQCs, than those potentially related to PQCs (starting from Shor's algorithm to the hidden linear functions of Bravyi-Gosset-Koenig). Second, saying that vanishing gradients are crucial is also false, since this dismisses the possibility of heuristics which do not see the barren plateaus in the training since these might exploit the structure or the information of the problem (as in the case of VQE for chemistry with tailored ansatz instead of using hardware efficient ansatz). Again, this is again a very narrow view of the authors to highlight the importance of the results by claiming domains not covered by the results in the paper which are limited to training approaches which rely on gradients, and with specific cost functions and with an specific proposed ansatz, which to the best of our knowledge has never been used elsewhere but in this paper. As mentioned, quantum speedup goes way beyond the training of PQCs with gradients and beyond the assumptions by the authors. This should be clearly stated if the comment on the exponential samples is to hold. As commented in our comments on the "trainability" term above, the most important information missing in all of this is an honest statement of the scope of the proofs in the paper.

4) Alternating layered ansatz (ALA) referred as a hardware efficient ansatz (HEA):

It is claimed in the new cover letter to the editors, the reply to other referees, and in the new version of the paper that the ALA proposed in this paper is a form of HEA, as a highlight of the paper, since it resonates with NISQ algorithms which are mostly tailored in the HEA paradigm. This is another statement which is far from true given that the construction of the ALA proposed in this work is against the basic principle behind a HEA proposal. To the best of our knowledge, the hardware-efficient ansatz was proposed in "Hardware-efficient variational quantum eigensolver for small molecules and quantum magnets. A. Kandala et al, Nature, 549, 242–246(2017)". As stated there, the hardware-efficient ansatz is efficient because it uses native entangling gates in the hardware in alternation with single-qubit gates. This has been the quantum algorithm community and every single paper pointing to a HEA construction refers to this construction, which very compactly applies this anternation of single-qubit and native two-qubit gates. In the ansatz proposed in this paper, and as a requirement for the main proofs to be sound in this work, the ALA requires 2-design in each of the m-qubit blocks of the ALA. As pointed by the authors, such 2-design unitaries need to be constructed out of native gates, and there is a significant overhead which scales with the number m of qubits in each block. By construction, the ALA construction is hardware-inefficient and we are not aware of any experimental or theoretical work beyond this paper which sees value in constructing the gates in this fashion. The story is different in the original paper on barren plateau which had similar requirements on the 2-design. But in that paper, the HEA ansatz was used as the RPQC and it was clear that to be within the margins of the proof, the circuit needed to have enough depth for it. We don't foresee any researcher attempting to construct the design principles of the ALA proposal here, since as the authors know it is

inefficient. As a proof of this point and its inefficiency, in the demonstration of the quantum autoencoder in this paper, even they avoided their own ALA design. This ALA design is not something we foresee anyone in the community would like to try experimentally other than for theoretical bounds as used here (and even less now after the results obtained by the authors).

Also the authors claim in the reply to other referees that the ansatz used in both of the recent Google papers (the Hartree-Fock VQE paper and the QAOA). Because of the arguments above, it is wrong to call that ansatz as an ansatz similar to the ALA proposed here since the only similarity is that they are composed of layers. But the authors use this statement to justify their ansatz and this is very misleading. For example, the main point used and what makes the ALA useful for the results proven in the paper is the 2-design of the m-qubit blocks. And none of the Google papers and to the best of our knowledge any other experimental or theoretical paper requires this constraint. We as reviewers are fine with the ALA as a theoretical construction to build the proof, but all misleading comments referring to this ansatz as a hardware-efficient or similar to other experiments should be removed or clarified to mention they are far from being comparable since they live at the opposite ends in their design principles.

For the most part the text was carefully worded in such a way that there is no conflict with the difference between HEA and ALA, since it is true that current hardware could be used to implement either the 1D or 2D ALA ansatz. The exception would be the new paragraph in the Discussion where it is discussed that layers can be added to a HEA (paragraph after the BK versus JW transformations), while nothing can be said about HEA given that the paper is about ALA. Clearly from the reply to referees and to the editors, the authors push this idea that HEA is similar to ALA, where as explained above, the HEA could not be more different since HEA is tailored for practically implementable while no one would attempt to use the ALA since it is not efficient given the added overhead in entangling gates to make it 2-design. This speaks of the design behind them (one for experiments and the other tailored to make progress in the mathematical proofs) and the authors need to acknowledge and point this in the paper since as it is, it is very misleading to the reader.

5) Claims from toy model quantum autoencoder:

The authors claim in the reply letter as well in the main body of the manuscript that the simulations they performed in the quantum autoencoder example is enough to illustrate that the numerical results seem to point that the mathematical proofs in the non-hardware-efficient ALA could be valid for the case of HEA like the one they tried in simulations. We appreciate the authors clearly stating that the HEA ansatz used does not meet the requirement for the proofs based on the ALA. But it is not correct to mention that these numerics seem to point that the proof is valid beyond the assumption of the ALA. Given the contrived and trivial nature of the specific construction of the example, this could easily be far from reality and it is not obvious it will hold for interesting and realistic examples. It is an interesting observation though, and we are not suggesting removing it. But the authors should leave it as it is: a numerical simulation on a trivial example that happens to exhibit a dependence on the cost function. The authors should clearly state as well that the dependence or performance of solvers in HEA is an open question and beyond the scope of the theorems. The same applies to the comments about black-box optimization versus gradient-based. The authors used the only trivial example that they run to make the big claim that their result seems to impact any solver, whether it is gradient-based or gradient-free. If one goes by the proofs in the paper, calculating the gradient seems to be the bottleneck and there are tens of gradient-free solvers and heuristics which don't rely on gradient estimation and that could be immune to these results. Therefore, instead of pushing for claiming that their methods seem to apply to other solvers, the authors should make a statement that the dependence on the solver and the heuristics is beyond the scope of this work.

Reviewer #3:

Remarks to the Author:

All of my questions were answered by the authors and they modified the manuscript accordingly. There still remains a question about the proof of Theorem 2, which the authors answered by showing examples in (85). So it is now reasonable to believe that the claim is true. By the way, there is a type in (85). So, I believe that the manuscript is ready for publication after the authors have fixed typos.

Dear Editors and Referees,

We again sincerely appreciate the helpful comments and suggestions made by the referees. Based on these comments we have revised our manuscript, and we believe that the changes should make our work much clearer. Below we provide a summary of the main changes:

- Following the suggestion from Referee 2 we have decided to change the the title of our manuscript. The title is now “Cost-Function-Dependent Barren Plateaus in Shallow Parametrized Quantum Circuits”.
- To make our work clearer we have made minor modifications to the abstract and introduction. Specifically, in the introduction we have re-arranged the order of the paragraphs so that the justification of the term “trainable” comes before its use, and we now clearly state our definition for this term.
- We have added relevant references to works showing that barren plateaus affect both derivative-based and derivative-free optimization methods.
- Based on the comments of Referee 1 we have improved Proposition 1. In our new result, the proof for the global cost C_G is valid for any $\delta \in (0, 1)$. The proof is in the revised Supplementary Information.
- We have clarified the text to avoid confusions as to how our results are relevant for 1D and 2D connectivity.
- To address an issue raised by Referee 2, we have clarified the name of our ansatz, and we now refer to it as an Alternating Layered Ansatz, rather than a layered hardware-efficient ansatz.
- To address issues raised by Referee 1, we now state in our main Theorems that our results are valid for the case when W_A , W_B and each block in the ansatz form independent two-designs.
- Following comments by Referee 2, we have now modified the text to clarify the connection between our numerics and our main results.
- We have added a paragraph in the Discussion section about heuristic methods to mitigate the effect of barren plateaus.
- We have corrected the typos in equations and text pointed out by the Referees 1 and 3, plus additional ones we found while revising our work.

In view of these improvements to our paper, we are resubmitting our manuscript “Cost-Function-Dependent Barren Plateaus in Shallow Quantum Neural Networks” to *Nature Communications*.

We additionally remark that our work has recently received significant attention from the community. This has been reflected by both the number of citations received (45 on google scholar) and the number of invited talks we have given for these results (which are readily available online). Moreover, we believe that our work has revived interest in the topic of barren plateaus, as can be seen by multiple recent preprints on this topic, several of which are specifically based on the present manuscript.

We appreciate the consideration of our manuscript, and the helpful feedback from the referees.

Sincerely,

Marco Cerezo, Akira Sone, Tyler Volkoff, Lukasz Cincio, and Patrick Coles

Our replies to the referee comments are below. The referee comments are quoted in blue italics.

I. Reply to Reviewer 1

I thank the authors for addressing the points raised. Due to time constraints I wasn't able to carefully proofread all the technical details in the Supplemental Material, but I do have some additional followup concerns/questions. Besides the following concerns, I am happy with the current version of the manuscript.

We thank the referee for their useful comments. Below we address the points raised by the referee.

1. The authors have fixed a bug in the previous version of the manuscript which has changed Proposition 1, which analyses the "narrow gorge" phenomenon for a toy problem. However, it seems to me that the current version of Prop. 1 is quite weak. The current version shows that with high probability, the global cost function is delta-far from the optimum in cost function value, for $\delta = 1 - (1 - c/n)^{1/n}$ with $c = O(1)$. However, asymptotically this scales like $\sim c/n^2$ which vanishes as n increases.

The previously submitted version of Proposition 1 contained a typo in the statement of the proposition, namely, the condition on δ should have been printed $\delta \leq 1 - (1 - \frac{c^2}{n})^n$, not $\delta \leq 1 - (1 - \frac{c^2}{n})^{1/n}$. However, this issue is now a moot point for our resubmitted manuscript. This is because, in our resubmission, we have improved the statement of Proposition 1 so that it now holds for all $\delta \in (0, 1)$.

So for example, Prop. 1 doesn't even rule out the possibility that the global cost function asymptotically approaches the optimum (like $\sim 1/n$ for example) with high probability when randomly initialized. Based on the explanation of the narrow gorge phenomenon, I would expect that with overwhelming probability, the global cost function value will be close to 1, which is much stronger than what Prop. 1 shows. Conversely, for the local cost function C_L , it is shown that the cost function is delta-close to the optimum with high probability if $\delta > 1/2$. Hence, the Prop. fails to prove that C_G and C_L behave much differently for this toy problem. For example, as far as I can tell the following possibility is not ruled out by this result: the distributions of C_L and C_G may be asymptotically identical as $n \rightarrow \infty$, concentrating around the value $C_L = C_G = 1/4$. Since the purpose of the Prop. is to establish significantly different behavior of C_L and C_G , I feel the result should be strengthened to rule out the sort of example above.

The referee's expectation is correct and, indeed, one way to show this behavior is to simply apply a basic expectation bound to the narrow gorge volume. In the new version of Proposition 1, we have done this. One concludes from Proposition 1 that for any $0 \leq \delta < 1$, the probability that C_G is less than δ decays exponentially with n . The part of Proposition 1 concerning C_L , which remains unchanged, shows that there is a neighborhood of the optimum that has finite probability asymptotically. So the "narrow gorge" terminology is now more apt. In particular, C_G cannot concentrate at a value less than 1.

2. I'm confused about the extension of the results to 2D. Referring to Fig. 3c, from the diagram it looks like the extension of the ansatz to 2D simply consists of grouping sites together to effectively collapse the 2D grid into a 1D chain, which can subsequently be analyzed as in the original 1D case. Indeed, you say "we remark that the following results are valid for both cases as the light-cone structure will be the same". However, I am struggling to understand how this picture is motivated. In a 2D $\sqrt{(n)} \times \sqrt{(n)}$ square grid of qubits, each block would then need to consist of at least $\sqrt{(n)}$ qubits to collapse to the 1D case, which does not seem well motivated, since each block needs to form a 2-design. Also, the lower bound of Theorem 2 becomes uninteresting if $m \sim \sqrt{(n)}$. It seems like the more natural setting in the 2D case would involve each block involving only a constant number of sites, and the blocks would be tiled in such a way to form a natural circuit architecture. However, in this setting the lightcone structure is not obviously equivalent to the 1D case, so I don't see how the 1D arguments are valid. Perhaps I'm just confused about your argument, in which case some extra discussion about how the 2D case reduces to the 1D case may be helpful. If you have in mind that one of the dimensions of the 2D grid be held constant so that $m = O(1)$, then I feel this should not be referred to as a 2D result as it effectively is still a 1D ansatz (it is just a way of implementing the 1D Alternating Layered Ansatz). Indeed, in this case I don't understand what the purpose is of drawing a distinction between 1D and 2D implementations of the (ultimately 1D) Alternating Layered Ansatz.

As the referee accurately points out in our manuscript we intended to say that the one-dimensional ansatz can be effectively created with 1D and 2D connectivity. The main advantage of using a 2D connectivity is that, as stated in the main text, the circuit depth required to make each block a 2-design is reduced.

Indeed, the light-cone structure would change in a two-dimensional alternating layered ansatz, and this would fall beyond the scope of our main results. To clarify this point we have added to the main text the following sentence in the ansatz section: "That is, with both one- and two-dimensional hardware connectivity one can group qubits to form an Alternating Layered Ansatz as in Fig. 3(a)". In addition, to avoid confusion we have removed the terms "1D and 2D" from our main theorems.

3. I think Theorem 2 (the lower bound) is either technically incorrect as stated or there is some notational ambiguity, as there appears to be a simple counterexample. Consider the circuit consisting of just one block of the form $\exp(-i\theta P)U$ which forms a 2-design, where P is a Pauli and U consists of some other unitaries. Suppose the target

operator O is simply $O = P$. Then the cost function is $C(\theta) = \langle 0|U^\dagger \exp(i\theta P)P \exp(-i\theta P)U|0 \rangle = \langle 0|U^\dagger P U|0 \rangle$. Clearly this is independent of θ , so the gradient of C is zero everywhere (and hence the variance is zero everywhere). However, Theorem 2 as written predicts a positive variance. Presumably this is just some sort of edge case which can be easily accounted for, but Theorem 2 should be somehow modified to rule out this counterexample (and other possible edge cases). This will happen more generally when we compute the derivative with respect to a rotation in the last layer which commutes with the operator O . Maybe Thm. 2 is not applicable for the last layer of blocks?

Here we remark that the example that the referee points out falls beyond the scope of our Theorem. This is due to the fact that our results are derived using the formula for the variance in Eq (31), which is valid when W_A and W_B in Eq (7) form independent 2-designs. Here we recall that the trainable parameter θ_μ is in a block W which can be expressed as $W = W_A W_B$, where θ_μ parametrized the first unitary in W_B (see Eq (30)). In the example that the referee presented W_A would correspond to the identity, as θ_μ parametrizes the first gate in W . Hence this case falls outside the scope of our theorems. Note that if one only assumes that either W_A (or W_B) forms a local 2-design the formulas become much more intricate and much more difficult to solve analytically. This is due to the fact that the integration over W_B (or W_A) cannot be symbolically performed.

To clarify this point we have added in our theorems the fact that the results are valid when W_A , W_B , and every block in V form independent 2-designs.

- page 1. "Ideally, in order to reduces" -> "reduce"
- page 3. "L layers of m -qubit gates". Should "gates" actually be something like "unitaries" since each block will generally consist of multiple individual gates?
- Fig. 3b. Missing a prime on O_i .
- page 5. "becomes trivial when the number of layers L is $O(\text{poly}(n))$ ". Should this technically be something like $\Omega(\text{poly}(n))$ or just $\text{poly}(n)$, since $O(\text{poly}(n))$ would generally be understood to include $\text{polylog}(n)$ for which your expressions are non-trivial?
- page 6. "dimensionality" -> dimensionality
- page 11. "later" -> latter
- page 13. Missing period at end of Section VII.

We thank the referee for point out these errors. We have fixed them in our current manuscript.

II. Reply to Reviewer 2

Report to the authors:

We feel the authors need to tone down and add clarity in the manuscript to all the major points raised in this report before it is considered further for publication. In the current form, the manuscript is still very misleading to the scientific community interested in NISQ algorithms.

We appreciate the sincere feedback from the referee, which helps to strengthen our paper. We have made a considerable effort to account for the feedback and consequently clarify our paper. Below we present a detailed description of the changes we have made to our manuscript.

Major revisions suggested:

1) *Use of QNNs in the body of text and title:*

In their reply the authors provide four arguments of why using the terminology of QNNs is appropriate and interchangeable with the notion of parametrized quantum circuits. We strongly disagree with most of them and it looks the authors feed the confusion of what a machine learning task is about and ignore the minimum requirement for what constitutes a neural network, either classical or quantum, independently of the many quantum proposals out there. To address the comments from the authors, we paste here their reply for convenience:

"First, note that it is now standard for the community to use the term QNN when referring to Parametrized Quantum Circuits (PQCs). For instance, the Nature Communications article in [1] employs the term QNN when working with PQCs. Hence, we are simply employing commonly used terminology. Second, using QNN can also be justified from the fact that neuron activation functions can be embedded inside unitary gates (see [2]). Third, note that the main numerical implementation performed in our manuscript was for quantum autoencoders, and it is standard practice

to view such autoencoders as neural networks. Because this is our main illustrative example, we wanted to draw the attention of the quantum machine learning community to our manuscript with the term QNN. Finally, as shown in our recent work [3], the results obtained in our current manuscript can be directly applied to other architectures that have already been called Quantum Neural Networks in the literature (see [4]). Hence, we believe that it is appropriate to employ the term QNN in our manuscript.”

These arguments are flawed in many ways and indicate a lack of understanding from the authors about even what is machine learning and what quantum neural networks are used for. Although some QNNs happen to be PQCs (the reason why the authors tend to believe the community is using the term QNN for PQCs), not all PQCs are QNNs. It is important to note that all or most of the papers that have used the term of QNN to denote PQCs are used in the context of machine learning tasks. For a learning task to be valid, one needs to have the concept of a training data set from which a learning task can be defined in the form of a cost function that measures the performance of a trained neural network. Examples of these QNNs based in PQCs are abundant and range from classification/supervised learning tasks and generative unsupervised learning tasks. Referring that the community is now using QNNs to refer to PQCs is misleading as it reinforces a confusion between distinct concepts. This is the basis of us requesting the authors change the title to be more precise to what their results are about. Since the authors themselves seem to want to sell their results as QNNs, it is fine to mention the connection of PQCs to QNNs inside the text, but they need to clarify this distinction.

It is interesting that in their argument they refer to the paper of McClean et al. published in Nat. Comms (Ref. [1]) and where the first results about barren plateaus were presented. Although it was used by the authors in the title, in reality QNNs are not mentioned anywhere in the text. We counted 1 occurrence on the work QNN in that paper, while about 100 times they used the expression random parametrized quantum circuits (RPQC). We feel that RPQC is a good description of what the theory inside this paper from Cerezo et al. is about, and that even using that term is generous since the theorems and results of this paper under review deals with a narrower scope: that of the layered version of the RPQC which requires a 2-design per block units in the layers. We also note a recent publication from the same group of authors here, <https://arxiv.org/abs/2007.14384>, use the term VQA in the title and avoid QNN. We feel that is already a more sensible term to be used in the title than QNN.

To summarize. Although some PQCs have been used to construct some QNNs proposals, it is important these are taken into the right context of learning tasks and not used interchangeably as the authors argue and use it in the text (for example, ground state preparation in chemistry or QAOA are not learning tasks since there is not a training set. But these both are VQA and PQC). As long as the PQC is used for a learning task, then QNN is a good term to be used there. Therefore, although the results here would be applied to that family of QNNs or some related ones which are based in layered RPQCs, saying that QNN are PQC are being used interchangeably in the community is wrong and unjustified. Any paper using QNN, if a decent one, at a minimum is solving either a supervised, unsupervised or reinforcement learning task. If this abuse of language has been used by some authors, this practice needs to stop for scientific integrity and correctness.

In view of the above comments from the referee, we have decided to change the title of our manuscript from “Cost-Function-Dependent Barren Plateaus in Shallow Quantum Neural Networks” to “Cost-Function-Dependent Barren Plateaus in Shallow Parametrized Quantum Circuits”.

The motivation for this change is simply for clarity. Using the more precise language of PQCs will precisely connect our title to the statements in our theorems. In addition, we have modified the abstract and the main text of our paper, emphasizing PQCs more and de-emphasizing QNNs. We hope these changes, both to the title and to the text, addresses the concerns about clarity that the referee raises.

With this stated, we nevertheless feel obligated to point out that our results do have relevance to certain Quantum Neural Networks (QNNs). We therefore felt that the referee’s comments were somewhat unwarranted, even though we are willing to make the aforementioned concessions in the interest of clarity.

The most obvious direct connection of our results to QNNs comes in our numerical implementation for the quantum autoencoder. The quantum autoencoder is literally a QNN. The quantum autoencoder can be viewed as a quantum generalization of the classical autoencoder, which is a type of artificial neural network. Note that the quantum autoencoder employs a training data set, and the cost function is defined as an average fidelity over training data points. Here, the training data (in the form of a summation in Eq (21) over input states) can always be absorbed into a VQA acting on a single input state (the statistical mixture of the data in the training set). This establishes a formal mathematical connection between a QNN with training data and a VQA with a single input state. Hence, the quantum autoencoder shows that the scope of our results are valid for quantum machine learning implementations using QNNs that are PQCs. In summary, our main numerical implementation is a QNN, and examination of the cost function for this example shows that for some quantum machine learning tasks, there is a mathematical connection between the training set perspective and the single input state perspective.

In addition, we remark that our theorems are valid for QNNs that employ a layered PQC and that have a cost function of the form (6). Again, we remark that, in some cases, cost functions based on training data can sometimes

be rewritten in the form of (6).

2) Usage of terms such as trainability and trainability guarantees:

The authors argue that they added a paragraph to justify the term trainability in the introduction. First, it is important to note the paragraph as written is misleading (see section below on quantum speedup for more details).

In the new version of our manuscript, we have clarified the paragraph which justifies the use of the term trainability. This paragraph no longer talks about quantum speedup, and now reads as follows:

“We remark that polynomially vanishing gradients imply that the number of shots needed to estimate the gradient should grow as $\mathcal{O}(\text{poly}(n))$. In contrast, exponentially vanishing gradients (i.e., barren plateaus) imply that derivative-based optimization will have exponential scaling [30], and this scaling can also apply to derivative-free optimization [31]. Assuming a polynomial number of shots per optimization step, one will be able to resolve against finite sampling noise and train the parameters if the gradients vanish polynomially. Hence, we employ the term “trainable” for polynomially vanishing gradients.”

Aside from the need for that to be addressed and corrected, the paragraph is the last one in the introduction while the authors refer to trainability several times before in the abstract and introduction. The definition of what is meant by trainability and trainability guarantees should be stated up front the first time the term is used. This is critical to the readability of the paper given that the definition used by the authors is unique and only applies to this paper.

The aforementioned paragraph has been moved in the introduction so that it now appears prior to any use of the term “trainability” in the main text. This should help to improve the readability of the manuscript.

As we stated in our first report, saying a PQC with global cost function is not trainable is misleading since there are heuristic strategies which might be successful and therefore disproving the trainability claims.

We agree with the referee that the language should be clarified in the manuscript. In its present form, we have removed the use of the term “untrainable” while only keeping the term trainable when the gradient vanishes (at most) polynomially with the system size. This change can be seen for instance in the sentence “However, here we argue that this cost function and others like it are untrainable” as it now reads “However, here we argue that this cost function and others like it exhibit exponentially vanishing gradients.”

Furthermore, instead of deviating the attention to claims on quantum speedup, which are secondary, the most important statement in the introduction and in the definition is to remark the definition itself. Basically, a clear statement that trainability as used in the paper is latched to the main assumptions in the proofs: 1) random initialization of parameters, 2) the specific details of the layered ansatz proposed which is not hardware-efficient (more on this below), and 3) the explicit form of the cost function which does not necessarily capture all cost functions used in the literature (e.g., in several QML papers the KL divergence is widely used and it not of the form proposed since it has a logarithm of the probability. Since the proofs don't mention these cases, it should be stated that the results apply to a family of local and global cost functions, but the most generic case of arbitrary cost function is not covered (or if the proofs cover any possible loss or cost function it would be to state that as well). We feel for the scientific integrity of the results presented, the authors should make an effort to state that saying anything beyond the assumptions in the proofs concerning this paper is beyond the scope and left as open questions. And this should be clear from early in the introduction before big claims about trainability are presented, and without clearly defining what is meant by that.

We would like to remark that, in the abstract, main text and conclusions we state that our results are valid for random layered ansatz and for cost functions of the form in Eq. (6). In fact, in the General Framework section we clearly state what is the scope of our results. Moreover, in our discussions we have a paragraph describing the limitations of our theorems. However, to further emphasize this point we have modified the first sentence of that paragraph in the discussions. It now reads: “We emphasize that while our theorems are stated for a hardware-efficient ansatz and for costs that are of the form (6), it remains an interesting open question as to whether other ansatzes, cost function, and architectures exhibit similar scaling behavior as that stated in our theorems.”

It is a good point to remark that there are proposals to bypass these limitations, as the case of the proposals to deal with parameter initialization. We feel this is a more honest and needed clarification since as written now, it looks like the authors' claims on the trainability of PQCs, are more general than what they really are.

We note here that some of the proposals for initializing ansatzes and attempting to mitigate the barren plateaus phenomena are heuristic methods which have no provable guarantees. In fact, strategies for parameter initialization

usually require knowledge about the problem at hand, which is not always available. While we believe that some of these methods are promising, we also think that more works needs to be done to come up with generic strategies can help in mitigating or even preventing the effect of barren plateaus.

To emphasize this point we have added a new paragraph at the end of the Discussion section which reads “Finally, we remark that some strategies have been developed to mitigate the effects of barren plateaus [30,31,51,52]. While these methods are promising and have been shown to work in certain cases, they are still heuristic methods with no provable guarantees that they can work in generic scenarios. Hence, we believe that more work needs to be done to better understand how to prevent, avoid, or mitigate the effects of barren plateaus. ”

3) New claims about quantum speedup:

As pointed out by the authors, they added a paragraph at the end of the introduction on quantum speedup. Again, the authors communicate this very poorly and in misleading way by stating “vanishing gradients are crucial for quantum speedup”. This is far from true for many reasons, First of all, there are more proofs on quantum speedup that are not even related to training PQCs, than those potentially related to PQCs (starting from Shor’s algorithm to the hidden linear functions of Bravyi-Gosset-Koenig). Second, saying that vanishing gradients are crucial is also false, since this dismisses the possibility of heuristics which do not see the barren plateaus in the training since these might exploit the structure or the information of the problem (as in the case of VQE for chemistry with tailored ansatz instead of using hardware efficient ansatz).

The sentence about quantum speedup was in the context of variational quantum algorithms, i.e. in the context of the present manuscript. Hence the context was not referring to non-variational quantum algorithms.

Nevertheless, to improve the clarity of our manuscript we have now removed the discussion about quantum speedup. We paste the new paragraph here for convenience, and one can see that quantum speedup does not appear here anymore:

“We remark that polynomially vanishing gradients imply that the number of shots needed to estimate the gradient should grow as $\mathcal{O}(\text{poly}(n))$. In contrast, exponentially vanishing gradients (i.e., barren plateaus) imply that derivative-based optimization will have exponential scaling [30], and this scaling can also apply to derivative-free optimization [31]. Assuming a polynomial number of shots per optimization step, one will be able to resolve against finite sampling noise and train the parameters if the gradients vanish polynomially. Hence, we employ the term “trainable” for polynomially vanishing gradients.”

Again, this is again a very narrow view of the authors to highlight the importance of the results by claiming domains not covered by the results in the paper which are limited to training approaches which rely on gradients, and with specific cost functions and with an specific proposed ansatz, which to the best of our knowledge has never been used elsewhere but in this paper. As mentioned, quantum speedup goes way beyond the training of PQCs with gradients and beyond the assumptions by the authors. This should be clearly stated if the comment on the exponential samples is to hold. As commented in our comments on the “trainability” term above, the most important information missing in all of this is an honest statement of the scope of the proofs in the paper.

We respectfully disagree with the referee on this particular point.

First we remark that the form of the cost function we analyze, i.e., $C = \text{Tr}[OV\rho V^\dagger]$, is quite general and widely used in many VQAs (see refs [11-29]), and also applies to certain Quantum Neural Networks that employ training data such as the quantum autoencoder example that we consider.

Second, the scope of our results goes beyond training methods that rely on gradients. In fact, we have recently uploaded a preprint (current reference [31]) where we show that barren plateaus affect both gradient-based and gradient-free optimization methods (see below in our reply for more details on this).

Third, we note below that the Alternating Layered Ansatz (ALA) that we consider is actually employed in recent state-of-the-art implementations for both chemistry and optimization applications (Refs. [7] and [8]). Here we are referring to two recent papers from the Google group, which employ the ALA. We elaborate on this point below.

4) Alternating layered ansatz (ALA) referred as a hardware efficient ansatz (HEA):

It is claimed in the new cover letter to the editors, the reply to other referees, and in the new version of the paper that the ALA proposed in this paper is a form of HEA, as a highlight of the paper, since it resonates with NISQ algorithms which are mostly tailored in the HEA paradigm. This is another statement which is far from true given that the construction of the ALA proposed in this work is against the basic principle behind a HEA proposal. To the best of our knowledge, the hardware-efficient ansatz was proposed in “Hardware-efficient variational quantum eigensolver for small molecules and quantum magnets. A. Kandala et al, Nature, 549, 242–246(2017)”. As stated there, the

hardware-efficient ansatz is efficient because it uses native entangling gates in the hardware in alternation with single-qubit gates. This has been the quantum algorithm community and every single paper pointing to a HEA construction refers to this construction, which very compactly applies this alternation of single-qubit and native two-qubit gates.

To improve the clarity of our work, we now simply refer to our ansatz as an alternating layered ansatz. This change was motivated from the fact that the the ALA simply assumes a given structure in how qubits are connected in each layer, while making no assumption on the gates in each block. This change can now be seen throughout the text, including our abstract, where we no longer use the term “layered hardware efficient layered ansatz”.

That being said, however, the ALA will be a Hardware Efficient Ansatz (HEA). To made this point clearer we have added this following sentence to the Ansatz section: “We remark that the Alternating Layered Ansatz will be a hardware-efficient ansatz so long as the gates that compose each block are taken from a set of gates native to a specific device.” In addition, the ALA can be considered as a special case of a HEA where the qubits in the quantum hardware have nearest-neighbor connectivity. In this case, one exploits the native connectivity to connect qubits instead of assuming that one can do large-range entangling gates.

In the ansatz proposed in this paper, and as a requirement for the main proofs to be sound in this work, the ALA requires 2-design in each of the m -qubit blocks of the ALA. As pointed by the authors, such 2-design unitaries need to be constructed out of native gates, and there is a significant overhead which scales with the number m of qubits in each block. By construction, the ALA construction is hardware-inefficient and we are not aware of any experimental or theoretical work beyond this paper which sees value in constructing the gates in this fashion. The story is different in the original paper on barren plateau which had similar requirements on the 2-design. But in that paper, the HEA ansatz was used as the RPQC and it was clear that to be within the margins of the proof, the circuit needed to have enough depth for it. We don't foresee any researcher attempting to construct the design principles of the ALA proposal here, since as the authors know it is inefficient.

While it is correct that for each block in the ALA to be a hardware efficient ansatz one has to construct it with native gates, this does not necessarily mean adding a significant overhead. As noted in the main text, each block can be a 2-design with a relatively shallow circuit depth, namely order $\mathcal{O}(m)$ depth. Moreover, for the common case of $m = 2$ one can always implement any 2-qubit gates using its optimal (fixed depth) decomposition. The latter is already expressed in terms of three entangling gates plus single qubit rotations, which are gates usually assumes in a hardware efficient ansatz.

Here the referee mentions that they don't foresee any researcher attempting to use this architecture. However, as stated in our previous reply, the ALA architecture was employed by the Google group in Refs [7] and [8], for chemistry and optimization applications, respectively. Specifically, as shown in [8], their ansatz is composed of gates acting on alternating pairs of qubits, and each 2-qubit gates is composed of three entangling gates plus single qubit rotations. Hence, both this ansatz and the one in our manuscript have the same depth for $m = 2$. While in the Google ansatz the single qubit are parametrized in a specific way that doesn't lead to each block being a 2-design, it is clear that this architecture can be employed to solve relevant problems.

We finally remark that in the original barren plateau paper one assumes that the unitary acting on n qubits forms a global 2-design, which usually requires order $\mathcal{O}(n)$ gates. This result is actually harder to implement (i.e., harder to achieve) than our random ALA with L layers with local 2-designs, as in this case the depth of the ansatz is $\mathcal{O}(mL)$. Hence for fixed m and sub-linear depth L , one can implement the ALA ansatz more efficiently than the ansatz in the original barren plateau paper.

As a proof of this point and its inefficiency, in the demonstration of the quantum autoencoder in this paper, even they avoided their own ALA design. This ALA design is not something we foresee anyone in the community would like to try experimentally other than for theoretical bounds as used here (and even less now after the results obtained by the authors).

As stated in our main text, the choice of ansatz of our numerics was motivated from the fact that one can observe the cost-function-dependent barren plateau phenomenon when using an ansatz that does not fall within the limits of our theorems. Specifically, here each block of the ALA does not form an exact 2-design, and yet one can find that the global cost is untrainable (see reply to point 5 below for additional details on this issue).

Also the authors claim in the reply to other referees that the ansatz used in both of the recent Google papers (the Hartree-Fock VQE paper and the QAOA). Because of the arguments above, it is wrong to call that ansatz as an ansatz similar to the ALA proposed here since the only similarity is that they are composed of layers. But the authors use this statement to justify their ansatz and this is very misleading. For example, the main point used and what makes

the ALA useful for the results proven in the paper is the 2-design of the m -qubit blocks. And none of the Google papers and to the best of our knowledge any other experimental or theoretical paper requires this constraint. We as reviewers are fine with the ALA as a theoretical construction to build the proof, but all misleading comments referring to this ansatz as a hardware-efficient or similar to other experiments should be removed or clarified to mention they are far from being comparable since they live at the opposite ends in their design principles.

As previously mentioned, the similarities in the ansatz employed here and by the Google group go beyond simply being composed of layers. In both cases, the ansatzes have nearest-neighbor connectivity, as this is usually the type of connectivity available in current superconducting qubit devices. Moreover, we again recall the ansatz used by the Google paper is composed of blocks of single qubit rotations interleaved with entangling gates. Such gate decomposition can allow for compilation of arbitrary 2-qubit gates, and hence can form a 2-design when adequately sampling the angles in the rotations. Hence, one can see that the ansatz employed in our manuscript is realistic and implementable, and is closely related to the ansatz implemented by the Google group in their recent papers.

For the most part the text was carefully worded in such a way that there is no conflict with the difference between HEA and ALA, since it is true that current hardware could be used to implement either the 1D or 2D ALA ansatz. The exception would be the new paragraph in the Discussion where it is discussed that layers can be added to a HEA (paragraph after the BK versus JW transformations), while nothing can be said about HEA given that the paper is about ALA. Clearly from the reply to referees and to the editors, the authors push this idea that HEA is similar to ALA, where as explained above, the HEA could not be more different since HEA is tailored for practically implementable while no one would attempt to use the ALA since it is not efficient given the added overhead in entangling gates to make it 2-design. This speaks of the design behind them (one for experiments and the other tailored to make progress in the mathematical proofs) and the authors need to acknowledge and point this in the paper since as it is, it is very misleading to the reader.

As previously mentioned, to clarify the scope of our results, we no longer mention the term hardware-efficient ansatz when referring to our results, but rather say that our work is valid for ALA with blocks forming 2-designs.

5) Claims from toy model quantum autoencoder:

The authors claim in the reply letter as well in the main body of the manuscript that the simulations they performed in the quantum autoencoder example is enough to illustrate that the numerical results seem to point that the mathematical proofs in the non-hardware-efficient ALA could be valid for the case of HEA like the one they tried in simulations. We appreciate the authors clearly stating that the HEA ansatz used does not meet the requirement for the proofs based on the ALA. But it is not correct to mention that these numerics seem to point that the proof is valid beyond the assumption of the ALA. Given the contrived and trivial nature of the specific construction of the example, this could easily be far from reality and it is not obvious it will hold for interesting and realistic examples. It is an interesting observation though, and we are not suggesting removing it. But the authors should leave it as it is: a numerical simulation on a trivial example that happens to exhibit a dependence on the cost function.

To clarify this issue, the final paragraph of the result section now reads “In summary, even though the ansatz employed in our heuristics is beyond the scope of our theorems, we still find cost-function-dependent barren plateaus, indicating that the cost-function dependent barren plateau phenomenon might be more general and go beyond our analytical results.”

In addition, in our Discussion section we now say that “Moreover, our numerics suggest that our theorems (which are stated for exact 2-designs) might be extendable in some form to ansatzes composed of simpler blocks, like approximate 2-designs [39].”

Here we decided to take the optimistic approach where the results observed in our numerics could indicate that the cost-function-dependent barren plateau phenomenon is more general than the scope of our theorems. This can motivate other researchers to try to generalize the results in our work, rather than saying that we simply happened to chose a problem which exhibit a dependence on the cost function.

The authors should clearly state as well that the dependence or performance of solvers in HEA is an open question and beyond the scope of the theorems. The same applies to the comments about black-box optimization versus gradient-based. The authors used the only trivial example that they run to make the big claim that their result seems to impact any solver, whether it is gradient-based or gradient-free. If one goes by the proofs in the paper, calculating the gradient seems to be the bottleneck and there are tens of gradient-free solvers and heuristics which don't rely on gradient estimation and that could be immune to these results. Therefore, instead of pushing for claiming that their methods seem to apply to other solvers, the authors should make a statement that the dependence on the solver and the heuristics is beyond the scope of this work.

Here we would like to point out that it is not correct to assume from the calculations of our manuscript that calculating the gradient seems to be the bottleneck. The statement about barren plateaus says that in average, the partial derivatives of the cost function will be exponentially small. However, this does not mean that optimization methods which don't rely on gradient estimation could be immune to these results.

In fact, in our recent preprint "Effect of barren plateaus on gradient-free optimization" (current Ref [31]) we rigorously show this not to be the case. Our main result in that work proves that cost function differences, which are the basis for making decisions in a gradient-free optimization, are exponentially suppressed in a barren plateau. Hence, without exponential precision, gradient-free optimizers will not make progress in the optimization. In addition to the analytical results, that preprint also shows numerics indicating that several popular gradient-free optimizers (Nelder-Mead, Powell, and Cobyła) exhibit exponential scaling when training inside a barren plateau.

With that being said, the results in our current manuscript do not rely on that preprint, as the gradient scaling results in our work do not rely on any particular optimization method.

III. Reply to Reviewer 3

All of my questions were answered by the authors and they modified the manuscript accordingly. There still remains a question about the proof of Theorem 2, which the authors answered by showing examples in (85). So it is now reasonable to believe that the claim is true. By the way, there is a typo in (85). So, I believe that the manuscript is ready for publication after the authors have fixed typos.

We thank the referee for their positive comments about the manuscripts. We have addressed the typo in Eq (85), and several other in the manuscript.

Reviewers' Comments:

Reviewer #1:

Remarks to the Author:

The authors have satisfactorily addressed my remaining questions and concerns from the prior round in the current version of the manuscript. I also think that the additional changes made to make the language more precise and the results more clear have significantly improved the manuscript. I'm therefore happy with the current version.

Reviewer #2:

Remarks to the Author:

The referees thank the authors for the careful revisions and changes. We believe the current version is much clear and accurate while still reflecting the important contributions of this work. We think the current version is now ready for publication in Nature Communication.

REVIEWERS' COMMENTS

Reviewer #1 (Remarks to the Author):

The authors have satisfactorily addressed my remaining questions and concerns from the prior round in the current version of the manuscript. I also think that the additional changes made to make the language more precise and the results more clear have significantly improved the manuscript. I'm therefore happy with the current version.

We again thank the Referee for their useful comments and suggestions.

Reviewer #2 (Remarks to the Author):

The referees thank the authors for the careful revisions and changes. We believe the current version is much clear and accurate while still reflecting the important contributions of this work. We think the current version is now ready for publication in Nature Communication.

We thanks the Referee for their comments and suggestions as we believe our manuscript is now clearer and ready for publication.